KA-TP-07-2024
MS-TP-24-08

# NNLL-fast 2.0: Coloured Sparticle Production at the LHC Run 3 with $\sqrt{S} = 13.6$ TeV

Wim Beenakker[a,*], Christoph Borschensky[b,†], Michael Krämer[c,‡], Anna Kulesza[d,§],
Eric Laenen[e,f,g,¶], Judita Mamužić[h,‖], Laura Moreno Valero[d,**]

[a] *Institute for Mathematics, Astrophysics and Particle Physics, Radboud University Nijmegen, Heyendaalseweg 135, Nijmegen, The Netherlands*

[b] *Institute for Theoretical Physics, Karlsruhe Institute of Technology, Wolfgang-Gaede-Str. 1, 76131 Karlsruhe, Germany*

[c] *Institute for Theoretical Particle Physics and Cosmology, RWTH Aachen University, Sommerfeldstr. 16, 52074 Aachen, Germany*

[d] *Institut für Theoretische Physik, University of Münster, Wilhelm-Klemm-Str. 9, 48149 Münster, Germany*

[e] *Institute of Physics/Institute for Theoretical Physics Amsterdam, University of Amsterdam, Science Park 904, 1098 XH Amsterdam, The Netherlands*

[f] *Nikhef, Theory Group, Science Park 105, 1098 XG, Amsterdam, The Netherlands*

[g] *Institute for Theoretical Physics, Utrecht University, Leuvenlaan 4, 3584 CE Utrecht, The Netherlands*

[h] *Institut de Física d'Altes Energies (IFAE), Edifici Cn, Campus UAB, 08193 Bellaterra (Barcelona), Spain*

---

[*] w.beenakker@science.ru.nl
[†] christoph.borschensky@kit.edu
[‡] mkraemer@physik.rwth-aachen.de
[§] anna.kulesza@uni-muenster.de
[¶] eric.laenen@nikhef.nl
[‖] judita.mamuzic@cern.ch
[**] l_more02@uni-muenster.de

**Abstract**

We report on updated precision predictions for total cross sections of coloured supersymmetric particle production at the LHC with a centre-of-mass energy of $\sqrt{S} = 13.6$ TeV, computed with the modern PDF4LHC21 set. The cross sections are calculated at an approximated NNLO accuracy in QCD and contain corrections from the threshold resummation of soft-gluon emission up to NNLL accuracy as well as Coulomb-gluon contributions including bound-state terms. The corrections are found to increase the cross sections and reduce the theoretical uncertainty as compared to the best available fixed-order calculations. These predictions constitute the state-of-the-art calculations and update the existing results for $\sqrt{S} = 13$ TeV. We make our new results publicly available in the version 2.0 update to the code package NNLL-FAST.

# 1 Introduction

The search for supersymmetry (SUSY) [1–6] is one of the most important objectives of the physics programme of the Large Hadron Collider (LHC). SUSY addresses the shortcomings of the Standard Model (SM) of particle physics in an elegant and compelling way. Consequently, over many decades, and in particular since the beginning of the LHC operations, there has been an immense interest of the community in the results of the searches. SUSY posits that each elementary particle in the SM is paired to a supersymmetric partner or sparticle − with squarks ($\tilde{q}$) and gluinos ($\tilde{g}$) being the superpartners of quarks and gluons, respectively. In the context of the Minimal Supersymmetric Standard Model (MSSM) with $R$-parity conservation [7,8], one of the most studied SUSY models, sparticles are always produced in pairs. Exactly such production processes have been and are currently being searched for at the LHC. On the basis of Run 1 and 2 data analyses at the ATLAS [9] and CMS [10] experiments the bounds on coloured sparticles masses of up to 1–1.9 TeV for squarks and 1.2–2.5 TeV for gluinos have been determined, with exact values depending on additional mass parameters of the electroweak SUSY sector and the examined search channel [11–24]. For the third generation squarks, i.e. the stops and sbottoms as superpartners of the top and bottom quarks, the experimental limits are a bit more relaxed, excluding stops and sbottoms up to masses of around 0.5–1.6 TeV, depending on the search channel [14, 19, 25–34].

One of the very important ingredients entering the experimental analysis and enabling an accurate derivation of the mass exclusion limits are precise theoretical predictions for the total cross sections for the processes of interest. The next-to-leading order (NLO) SUSY-QCD corrections to squark and gluino production, both for total production rates, decays, as well as differential distributions, have been calculated some time ago [35–45]. The electroweak NLO corrections are also known [46–53]. Due to the high mass limits, the kinematical region where squarks and gluinos are produced close to their production threshold is of increased importance, and a significant contribution to the total cross section comes from this region. Near threshold, additional hard gluon radiation is strongly suppressed, forcing the radiation to be soft. Soft radiation, in turn, brings about large logarithmic contributions to the cross sections, which need to be systematically taken into account. The summation of the soft-gluon contributions to all orders in the strong coupling constant $\alpha_s$ can be performed by means of threshold resummation techniques in Mellin-moment space [54–59]. Resummed results for squark and gluino production, including stops, were first obtained at the next-to-leading logarithmic (NLL) accuracy, both in the Mellin-space approach [60–67] and in the framework of soft collinear ef-

fective theory (SCET) [68–70]. The accuracy of resummation was later increased to the next-to-next-to-leading logarithmic (NNLL) level, again in both the Mellin-space approach [71–77] and in SCET [78–81]. Recently, in [82], soft-gluon corrections in the Mellin-space resummation formalism have also been calculated for squark production in a non-minimal SUSY model, the Minimal R-symmetric Supersymmetric Standard Model [83], and matched to the existing NLO-QCD corrections [84].

In this work, we report on updated predictions for the cross sections for squark and gluino production processes in the MSSM at the approximated next-to-next-to-leading-order (NNLO) matched to NNLL accuracy for LHC Run 3 with a collision energy of $\sqrt{S} = 13.6$ TeV. The $\mathrm{NNLO_{Approx}}$+NNLL results are the most precise theoretical predictions currently available, including also resummation of Coulomb contributions as well as corrections from bound-state formation in the final state. The results have been consistently used by both the ATLAS and CMS collaborations in the analyses of SUSY searches in Run 2. The predictions for Run 3, presented here, can be obtained with the version 2.0 of the publicly available package NNLL-FAST. They correspond to an update of the Run 2 predictions in [77], provided by earlier versions of the package, in line with the upgrade at the LHC Run 3. The two sets of predictions differ not only by the value of the centre-of-mass energy but also by the sets of parton distribution functions (PDFs) with which they are obtained. The aim of this paper, similarly to [66] for NLO+NLL calculations, is to provide in one document a brief overview of the results that can be obtained with NNLL-FAST 2.0 (central values of the cross sections, error estimates and the $K$-factors), together with the calculations that led to them, as well as to discuss the impact of the differences in the NNLL-FAST set-up on the predictions.

The paper is structured as follows. In Sec. 2 we introduce the production processes of interest for this work. In Sec. 3, we review the higher-order calculations and in particular the threshold resummation formalism at NNLL accuracy, and briefly discuss the treatment of the various uncertainties. The implementation and parameters used in the code package NNLL-FAST as well as the version 2.0 update is detailed in Sec. 4. Numerical results are presented in Sec. 5, where we also provide comparisons with results obtained using an earlier version of the NNLL-FAST code, and we conclude in Sec. 6.

## 2 Squark and gluino production at the LHC

The coloured sector of the MSSM consists of the superpartners of the quarks and gluons, the scalar squarks $\tilde{q}$ and the fermionic gluinos $\tilde{g}$, the latter of which are of Majorana fermionic nature. Due to the colour charge of squark and gluinos, their production cross

sections at the LHC are predicted to be large and dominate over cross sections for other supersymmetric particles. This results in the already mentioned relatively high exclusion limits on the squark and gluino masses (in comparison with exclusion limits on masses of other particles), established from Run 1 and 2 data. The corresponding experimental analysis relies often on certain simplified scenarios such as decoupling limits where all supersymmetric particles other than the ones that are searched for are assumed to be very heavy and therefore decoupled from the production process and out of reach for direct searches with current experiments.

Assuming $R$-parity conservation, supersymmetric particles can only be produced in pairs. In the following, we will discuss only the dominant SUSY-QCD production channels. For squarks and gluinos, the following inclusive pair production processes can take place at a hadron collider with two colliding hadrons $h_1$ and $h_2$ (where in the case of the LHC, $h_1$ and $h_2$ are both protons):

$$h_1 h_2 \rightarrow \tilde{g}\tilde{g},\, \tilde{q}\tilde{q}^*,\, \tilde{q}\tilde{g},\, \tilde{q}\tilde{q} + X\,, \tag{2.1}$$

where $X$ stands for any additional radiation. We label the four types of processes in the following as:

- $\tilde{g}\tilde{g}$: *gluino-pair* production,

- $\tilde{q}\tilde{q}^*$: *squark-antisquark* production,

- $\tilde{q}\tilde{g}$: *squark-gluino* production,

- $\tilde{q}\tilde{q}$: *squark-squark* or *squark-pair* production.

For the latter two processes, here and in the following, we always assume the charge-conjugated processes $h_1 h_2 \rightarrow \tilde{q}^*\tilde{g},\, \tilde{q}^*\tilde{q}^* + X$ to be implied[1], i.e. when e.g. referring to $\tilde{q}\tilde{q}$ production, we mean the sum of $\tilde{q}\tilde{q}$ and $\tilde{q}^*\tilde{q}^*$. At the partonic level, the following initial-state channels contribute to the production processes at leading order (LO):

$$q_i\bar{q}_i,\, gg \rightarrow \tilde{g}\tilde{g}\,, \qquad q_{i'}\bar{q}_{j'},\, gg \rightarrow \tilde{q}_i\tilde{q}_j^*\,, \qquad q_i g \rightarrow \tilde{q}_i\tilde{g}\,, \qquad q_i q_j \rightarrow \tilde{q}_i\tilde{q}_j\,, \tag{2.2}$$

as well as the charge-conjugated processes, whenever appropriate. Here, the (s)quark indices $i^{(\prime)}$ and $j^{(\prime)}$ denote the (s)quark flavour. While squark-antisquark production through the $gg$ initial-state channel is always flavour diagonal, squark-antisquark and

---

[1] Note that in the MSSM, as gluinos are Majorana fermions, they are their own antiparticles, such that we do not distinguish between $\tilde{g}$ and $\bar{\tilde{g}}$.

squark-pair final states include also squarks of different flavours $i \neq j$, produced through the $q\bar{q}$ and $qq$ initial-state channels, respectively. The tree-level Feynman diagrams for the partonic production channels in Eq. (2.2) are shown in Figure 1.

In Eq. (2.1), we sum over the two chirality states $\tilde{q}_L$, $\tilde{q}_R$ of the squarks[2]. We assume the superpartners of the light quarks $(u, d, c, s, b)$ to be mass degenerate, leading to a 10-fold squark degeneracy. Due to the absence of top quark densities in parton distribution functions (PDF), there are fewer diagrams for the production of superpartners of the top quark than for the production of supersymmetric partners of light quarks (which e.g. includes the second diagram with $t$-channel gluino exchange in Figure 1 (b)). Moreover, also in contrast to the light quark case, mixing effects of the left- and right-handed superpartners in the stop mass matrix cannot be neglected due to the large top quark masses. We thus consider stop-antistop production separately:

$$h_1 h_2 \to \tilde{t}_a \tilde{t}_a^* + X \qquad a = 1, 2 \,, \tag{2.3}$$

where $a = 1, 2$ conventionally denote the light and heavy states of the stop, respectively[3]. When appropriate, the same treatment can be applied also to the case of the superpartners of the bottom quark, the sbottoms, so that we consider their production separately, and thus only assume an 8-fold degeneracy of the superpartners $\tilde{q}$ of $(u, d, c, s)$.

The hadronic production cross section for squark and gluino production can be written as a convolution of PDFs and the partonic cross section:

$$\sigma_{h_1 h_2 \to kl}\big(\rho, \{m^2\}\big) = \sum_{i,j} \int dx_1 \, dx_2 \, d\hat{\rho} \, \delta\Big(\hat{\rho} - \frac{\rho}{x_1 x_2}\Big) \\ \times f_{i/h_1}(x_1, \mu^2) \, f_{j/h_2}(x_2, \mu^2) \, \sigma_{ij \to kl}\big(\hat{\rho}, \{m^2\}, \mu^2\big) \,, \tag{2.4}$$

with $k, l = \tilde{q}^{(*)}, \tilde{g}, \tilde{t}^{(*)}$ and $i, j = q, \bar{q}, g$. Here, the variable $\rho := (m_k + m_l)^2 / S$ is given by the ratio of the sum of the final-state masses $m_k$ and $m_l$ squared with respect to the hadronic squared centre-of-mass energy $S$. $\{m^2\}$ stands for all the masses (such as squark and gluino masses) entering the calculation. Furthermore, $f_{i/h_1}(x_1, \mu^2)$ and $f_{j/h_2}(x_2, \mu^2)$ denote the PDFs which can, at LO, be interpreted as probabilities for partons with

---

[2]As squarks are scalar particles, they cannot carry chirality. The labels $L$ and $R$ are only used to distinguish the superpartners of left- and right-handed quarks. We furthermore neglect the masses of the five light quarks other than the top quark, so that the $L$ and $R$ squark states correspond to their mass eigenstates.

[3]We do not consider mixed $\tilde{t}_1 \tilde{t}_2^*$ or same-charge $\tilde{t}_{1,2} \tilde{t}_{1,2}$ production, as these processes are strongly suppressed at tree-level by vanishing top quark PDFs and thus would have to be considered as loop-induced processes, which receive the usual suppression from small coupling constants and loop factors with respect to the above-mentioned tree-level processes.

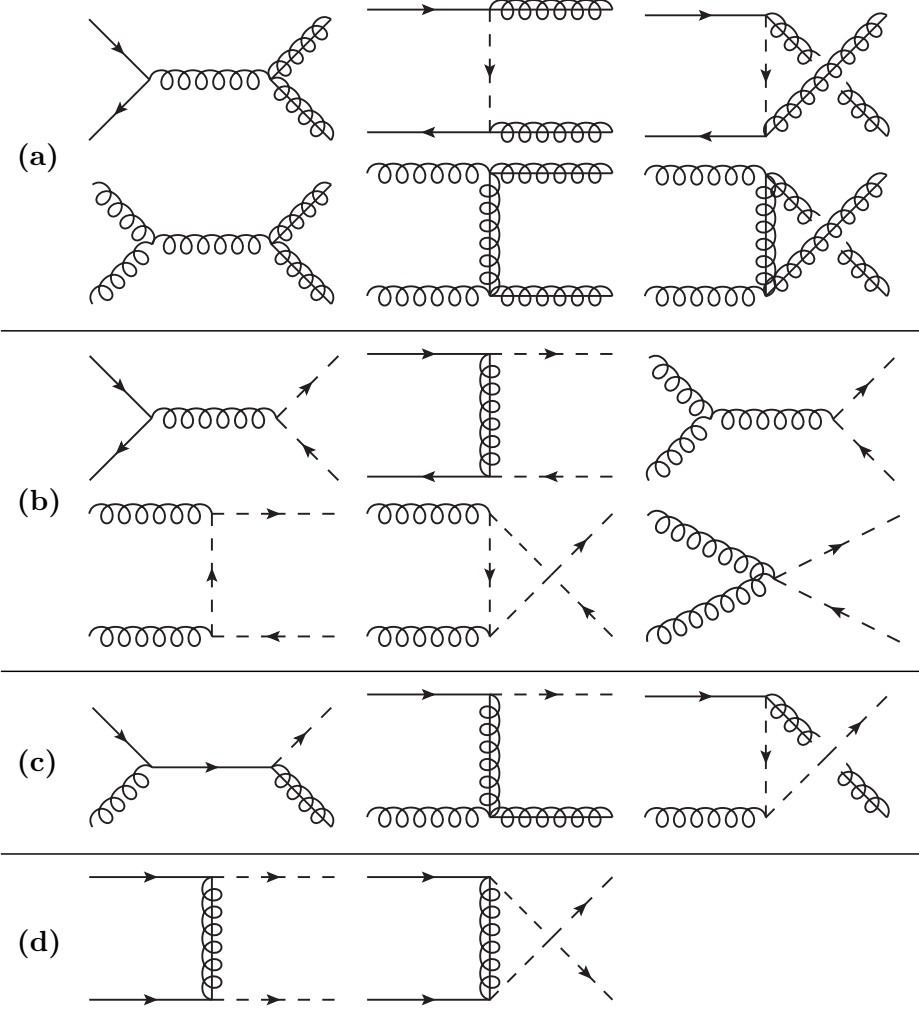

**Figure 1:** Tree-level Feynman diagrams for the partonic squark and gluino production processes of Eq. (2.2): (a) gluino-pair production $\tilde{g}\tilde{g}$, (b) squark-antisquark production $\tilde{q}\tilde{q}^*$, (c) squark-gluino production $\tilde{q}\tilde{g}$, (d) squark-pair production $\tilde{q}\tilde{q}$. Solid lines with an arrow correspond to quarks, curly lines to gluons, dashed lines with an arrow to squarks, and curly lines with a solid line in the middle to gluinos.

flavours $i$ and $j$ to be present inside the hadrons $h_1$ and $h_2$ and carrying momentum fractions $x_1$ and $x_2$, respectively, of the full hadronic momenta. $\sigma_{ij \to kl}$ stands for the partonic cross section. The scale $\mu$ corresponds to the factorisation scale $\mu_F$, separating long- and short-distance physics, which we set equal to the renormalisation scale in all of our calculations, $\mu := \mu_F = \mu_R$.

## 3 Higher-order calculations

Since the considered production processes are dominated by QCD and SUSY-QCD interactions, higher-order corrections can be sizeable. Therefore, they need to be taken into account in order to obtain reliable theoretical predictions and to reduce theoretical uncertainties. Below we briefly review the calculations for current state-of-the-art NNLO$_{\text{Approx}}$+NNLL predictions for inclusive cross sections for squark and gluino production at the LHC.

### 3.1 Fixed-order contribution

The fixed-order contribution NNLO$_{\text{Approx}}$ is an approximation of the NNLO SUSY-QCD result, consisting of $\sigma^{\text{NLO}}_{h_1 h_2 \to kl}$, the full NLO SUSY-QCD cross section at $\mathcal{O}(\alpha_s^3)$, and $\Delta\sigma^{\text{NNLO}_{\text{Approx}}}_{h_1 h_2 \to kl}$ which is an approximation of the $\mathcal{O}(\alpha_s^4)$ corrections:

$$\sigma^{\text{NNLO}_{\text{Approx}}}_{h_1 h_2 \to kl} = \sigma^{\text{NLO}}_{h_1 h_2 \to kl} + \Delta\sigma^{\text{NNLO}_{\text{Approx}}}_{h_1 h_2 \to kl} \,. \tag{3.1}$$

The inclusive NLO SUSY-QCD production cross sections for squarks and gluinos at hadron colliders have been calculated over 25 years ago [37, 38], and are implemented in the PROSPINO code [85], where in the more recent PROSPINO 2 version additional SUSY processes have been included. More recently, about 10 years ago, the calculation of squark and gluino production at NLO-QCD has been automated based on the MADGOLEM tool [40], and the squark-antisquark and squark-pair production processes have been recalculated at NLO-QCD including also decays and matching to parton showers, keeping all squark masses separate, i.e. without assuming a 10-fold squark degeneracy [42, 43]. Squark and gluino production can now be calculated, both for differential and total rates, in a fully automatised manner up to NLO-QCD using the MADGRAPH5_AMCNLO tool [45]. We refer the reader interested in the details of the NLO calculations to the original literature.

The $\Delta\sigma^{\text{NNLO}_{\text{Approx}}}_{h_1 h_2 \to kl}$ correction collects the $\mathcal{O}(\alpha_s^4)$ contributions which are enhanced in the limit of sparticle pair-production taking place close to the threshold, originating from

soft-gluon radiation, Coulomb-like emissions as well as two-loop non-Coulomb potential and kinetic-energy corrections, see [86]. Correspondingly, special care needs to be taken when matching $\Delta\sigma_{h_1h_2\to kl}^{\mathrm{NNLO_{Approx}}}$ and the full NLO correction to the resummed results, as discussed in the next section.

## 3.2 Threshold resummation

Due to the high exclusion limits, squarks and gluinos – should they exist – must be heavy, and the dominant contribution to their production cross sections stems from the threshold region where the sum of the final-state masses is close to the hadronic centre-of-mass energy, $S \to (m_k + m_l)^2$. In this limit, all additional radiation at higher orders is constrained to be soft, and the corrections due to soft-gluon emission have the general form

$$\alpha_{\mathrm{s}}^n \ln^m \beta^2 \,, \ m \le 2n \qquad \text{with} \qquad \beta^2 \equiv 1 - \hat{\rho} = 1 - \frac{4m_{\mathrm{av}}^2}{s} \,, \qquad (3.2)$$

where $m_{\mathrm{av}} := (m_k + m_l)/2$ is the average mass of the final-state particles $k$ and $l$, $s = x_1 x_2 S$ is the partonic centre-of-mass energy squared, and $\alpha_{\mathrm{s}}$ denotes the strong coupling. In the threshold limit, $\beta \to 0$, the logarithms of Eq. (3.2) become large and thus have to be taken into account at all orders not to spoil the perturbative expansion in $\alpha_{\mathrm{s}}$.

We carry out the all-order resummation of the soft-gluon emission after taking a Mellin transform of the hadronic cross section,

$$\begin{aligned}
\tilde{\sigma}_{h_1h_2\to kl}(N, \{m^2\}) &\equiv \int_0^1 d\rho \, \rho^{N-1} \, \sigma_{h_1h_2\to kl}(\rho, \{m^2\}) \\
&= \sum_{i,j} \tilde{f}_{i/h_1}(N+1, \mu^2) \, \tilde{f}_{j/h_2}(N+1, \mu^2) \, \tilde{\sigma}_{ij\to kl}(N, \{m^2\}, \mu^2) \,,
\end{aligned} \qquad (3.3)$$

where the threshold-enhanced terms are now of the form $\alpha_{\mathrm{s}}^n \log^m N$, $m \le 2n$, depending on the Mellin moments $N$, and the threshold limit is given by $N \to \infty$. The $\tilde{f}_{i/h_1}(N+1, \mu^2)$ denote the PDFs in Mellin space. The Mellin-transformed partonic cross section $\tilde{\sigma}_{ij\to kl}(N, \{m^2\}, \mu^2)$ then factorises into a product of terms separating hard and soft as well as soft-collinear contributions, allowing for a systematic reorganisation of the enhanced logarithms in terms of exponential functions [54–59]. The fully factorised result in terms of the resummed functions is then given as:

$$\begin{aligned}
\tilde{\sigma}_{ij\to kl}^{(\mathrm{res})}(N, \{m^2\}, \mu^2) = \sum_I \tilde{\sigma}_{ij\to kl,I}^{(0)}(N, \{m^2\}, \mu^2) \, C_{ij\to kl,I}(N, \{m^2\}, \mu^2) \\
\times \, \Delta_i \Delta_j \Delta_{ij\to kl,I}^{(\mathrm{s})}(N+1, Q^2, \mu^2) \,,
\end{aligned} \qquad (3.4)$$

where we introduced the hard scale $Q^2 = 4m_{\mathrm{av}}^2$, and where the cross section is split up into colour channels $I$ in an $s$-channel colour basis, in which the factorisation of soft and hard parts becomes diagonal [60,61,68], and $\Delta_i \Delta_j \Delta_{ij \to kl,I}^{(\mathrm{s})}$ are the functions containing the resummed threshold logarithms:

$$\Delta_i \Delta_j \Delta_{ij \to kl,I}^{(\mathrm{s})} \;=\; \exp\left[ L g_1(\alpha_{\mathrm{s}} L) + g_2(\alpha_{\mathrm{s}} L) + \alpha_{\mathrm{s}} g_3(\alpha_{\mathrm{s}} L) + \dots \right] \tag{3.5}$$

with $L := \ln N$. In Eq. (3.5), the perturbative series is now organised differently: while the exponential function takes into account terms up to all orders in $\alpha_{\mathrm{s}}$, the functions $g_1$, $g_2$, $g_3$ etc. now define different logarithmic orders of the approximation, with the first summand in the exponent $L g_1(\alpha_{\mathrm{s}} L)$ resumming terms up to leading-logarithmic (LL) accuracy, including additionally the second term $g_2(\alpha_{\mathrm{s}} L)$ denotes next-to-leading logarithmic (NLL) and including also the third term $\alpha_{\mathrm{s}} g_3(\alpha_{\mathrm{s}} L)$ denotes next-to-next-to-leading logarithmic (NNLL) accuracy of the threshold resummation. Expressions for the $g_1$, $g_2$, and $g_3$ functions can be found in e.g. [61,71].

In Eq. (3.4), the matching coefficient $C_{ij \to kl,I}(N, \{m^2\}, \mu^2)$ is given by:

$$C_{ij \to kl,I} = \mathcal{C}_{ij \to kl,I}^{\mathrm{Coulomb}} \times \left(1 + \frac{\alpha_{\mathrm{s}}}{\pi} \mathcal{C}_{ij \to kl,I}^{(1)} + \dots \right). \tag{3.6}$$

The factor $\mathcal{C}_{ij \to kl,I}^{\mathrm{Coulomb}}$ in Eq. (3.6) resums threshold-enhanced terms due to Coulomb-gluon exchange between slowly-moving final-state particles by employing the Coulomb Green's function of non-relativistic QCD with a NLO Coulomb potential, see [77] for more details. The $N$-independent terms originating from higher-order virtual contributions, excluding the Coulomb-gluon exchange between final states to avoid the double counting of these contributions, are collected by $\mathcal{C}_{ij \to kl,I}^{(1)}$.

In order to obtain physical results, the hadronic cross section in Mellin space must be transformed back to physical space. This is done by performing an inverse Mellin transform according to the minimal prescription [87]. In addition to the inverse Mellin transformation, we match the resummed cross section to the best available fixed-order calculation. To avoid the double counting of terms that occur in both the resummed as well as the fixed-order calculations, we expand the resummed cross section up to the available fixed order, which is of $\mathcal{O}(\alpha_{\mathrm{s}}^4)$ in our case, and subtract it. Then, we add the fixed-order cross section $\sigma_{h_1 h_2 \to kl}^{\mathrm{NNLO_{Approx}}}$ of Eq. (3.1). This methodology ensures that we combine the best known fixed-order result, covering also the kinematical region away from threshold, with the dominant threshold-enhanced corrections beyond the fixed

order, yielding:

$$\sigma_{h_1 h_2 \to kl}^{\text{NNLL-FAST}}(\rho, \{m^2\}, \mu^2) = \sigma_{h_1 h_2 \to kl}^{\text{BS}}(\rho) + \sigma_{h_1 h_2 \to kl}^{\text{NNLO}_{\text{Approx}}}(\rho, \{m^2\}, \mu^2)$$
$$+ \sum_{i,j} \int_{\text{CT}} \frac{dN}{2\pi i} \, \rho^{-N} \, \tilde{f}_{i/h_1}(N+1, \mu^2) \, \tilde{f}_{j/h_2}(N+1, \mu^2)$$
$$\times \left[ \tilde{\sigma}_{ij \to kl}^{(\text{res, NNLL, Coulomb})}(N, \{m^2\}, \mu^2) - \tilde{\sigma}_{ij \to kl}^{(\text{res, NNLL, Coulomb})}(N, \{m^2\}, \mu^2)\big|_{\text{NNLO}} \right].$$
$$(3.7)$$

As explained above, we perform simultaneous resummation of soft-gluon emission up to NNLL and of threshold-enhanced Coulomb contribution, denoted by the superscripts 'res, NNLL, Coulomb' in Eq. (3.7). In addition, corrections due to the formation of bound states between final-state particles are included in the $\sigma_{h_1 h_2 \to kl}^{\text{BS}}$ term. Again, we refer to the previous publication of [77] for more details on the calculation of this term. The threshold-enhanced two-loop contribution $\Delta\sigma_{h_1 h_2 \to kl}^{\text{NNLO}_{\text{Approx}}}$ as included in $\sigma_{h_1 h_2 \to kl}^{\text{NNLO}_{\text{Approx}}}$ differs from the corresponding term of the same order in the expansion of the resummed cross section above by subleading contributions that are suppressed in Mellin space as $\mathcal{O}(1/N)$. Our final result at NNLO$_{\text{Approx}}$+NNLL accuracy, denoting the state-of-the-art precision for predictions for squark and gluino production at the LHC, we label $\sigma_{h_1 h_2 \to kl}^{\text{NNLL-FAST}}$, which is implemented in the publicly available code package NNLL-FAST which we describe, in the context of the updates for the LHC Run 3 at $\sqrt{S} = 13.6$ TeV, below in Sec. 4.

### 3.3 Estimation of theoretical uncertainties

**Factorisation and renormalisation scale uncertainty**  As previously mentioned, we use for our calculations of both the fixed-order as well as the threshold-resummed cross sections a common factorisation and renormalisation scale $\mu$. We vary the scale $\mu$ around a central value chosen as the average mass of the final-state particles, $\mu_0 = m_{\text{av}}$, up and down by a factor of two[4], $\mu \in [\mu_0/2, 2\mu_0]$, to obtain an estimate of the remaining scale dependence and thus missing higher-order corrections. We determine the relative scale uncertainty with respect to the cross section evaluated at the central scale $\sigma_{\mu=\mu_0}$ as:

$$\delta_{\mu_0/2} := \frac{\sigma_{\mu=\mu_0/2} - \sigma_{\mu=\mu_0}}{\sigma_{\mu=\mu_0}}, \qquad \delta_{2\mu_0} := \frac{\sigma_{\mu=2\mu_0} - \sigma_{\mu=\mu_0}}{\sigma_{\mu=\mu_0}}. \qquad (3.8)$$

---

[4]In the calculation of the Coulomb-gluon and bound-state contributions, two additional characteristic scales appear, the Coulomb as well as the Bohr scale, which are different to the common factorisation and renormalisation scale $\mu$, see [77] for details. When varying $\mu$, we simultaneously vary the Coulomb and Bohr scales up and down by a factor of two. Thus, in the following uncertainties, a variation of these additional scales is implied.

The lower and upper bounds $\delta_{\mu-}$ and $\delta_{\mu+}$, respectively, of the scale uncertainty are then defined as follows:

$$\delta_{\mu+} := \max(\delta_{\mu_0/2}, \delta_{2\mu_0}), \qquad \delta_{\mu-} := \min(\delta_{\mu_0/2}, \delta_{2\mu_0}). \tag{3.9}$$

With this definition, the relation $\delta_{\mu-} \leq \delta_{\mu+}$ is always ensured. It is possible that both $\delta_{\mu-}$ and $\delta_{\mu+}$ have the same sign, in which case the bounds of the scale uncertainties should be calculated as

$$\delta_{\mu+,0} := \max(0, \delta_{\mu+}), \qquad \delta_{\mu-,0} := \min(\delta_{\mu-}, 0), \tag{3.10}$$

so that we always have $\delta_{\mu-,0} \leq 0 \leq \delta_{\mu+,0}$.

**PDF+$\alpha_{\mathrm{s}}$ uncertainty**   As a choice of parton densities, we make use of the recent PDF4LHC21 set [88], which combines the CT18 [89], MSHT20 [90], and NNPDF3.1 [91] sets in a global fit. The uncertainty associated with the procedure of generating the PDFs is encoded in separate eigenvector or replica sets in the case of a Hessian or a Monte Carlo representation of the set, respectively. Following the prescription in [88], they can then be used to determine an additional theoretical uncertainty on the cross sections from the PDF determination.

In our calculations, we use the Hessian set of PDF4LHC21 with one central and 40 eigenvector members as well as $\alpha_{\mathrm{s}}$ variation, `PDF4LHC21_40_pdfas`, where the requirement of positive-definite PDFs at high $x$ values has been imposed. The relative 68% C. L. PDF uncertainty according to the symmetric Hessian prescription is then obtained by computing the cross section for the central, $\sigma^{(0)}$, and each eigenvector set, $\sigma^{(i)}$ with $i = 1, \ldots, N_{\mathrm{set}}$ and $N_{\mathrm{set}} = 40$, as:

$$\delta_{\mathrm{PDF}} := \frac{1}{\sigma^{(0)}} \sqrt{\sum_{i=1}^{N_{\mathrm{set}}} \left(\sigma^{(i)} - \sigma^{(0)}\right)^2} \tag{3.11}$$

The PDF4LHC21 set includes additionally two members accounting for the 68% C. L. variation of $\alpha_{\mathrm{s}}$ around its central value $\alpha_{\mathrm{s}}(M_Z) = 0.118$ within the determination procedure of the PDFs. We use these members in our calculation to evaluate the $\alpha_{\mathrm{s}}$ uncertainty associated with the cross section around the central value $\sigma_{\alpha_{\mathrm{s}}(M_Z)=0.118}$. The relative $\alpha_{\mathrm{s}}$ uncertainty is then given as:

$$\delta_{\alpha_{\mathrm{s}}} := \frac{\sigma_{\alpha_{\mathrm{s}}(M_Z)=0.119} - \sigma_{\alpha_{\mathrm{s}}(M_Z)=0.117}}{2\sigma_{\alpha_{\mathrm{s}}(M_Z)=0.118}}. \tag{3.12}$$

The combined relative PDF+$\alpha_s$ uncertainty is then obtained as

$$\delta_{\text{PDF}+\alpha_{\text{s}}} := \sqrt{(\delta_{\text{PDF}})^2 + (\delta_{\alpha_{\text{s}}})^2}. \tag{3.13}$$

**Parametric uncertainty for $\tilde{t}_1\tilde{t}_1^*$ production**   In the case of stop-antistop production, the cross section depends, in addition to the mass of the produced stops $m_{\tilde{t}_1}$, on additional parameters such as the gluino mass $m_{\tilde{g}}$, the light-flavoured[5] squark mass $m_{\tilde{q}}$, as well as the mass of the heavier stop $m_{\tilde{t}_2}$ and the stop mixing angle $\theta_{\tilde{t}}$. However, the dependence on these additional parameters is suppressed, since they only appear as loop effects starting from NLO-QCD. We have checked that the dependence of the stop production cross section on $m_{\tilde{q}}$ and $m_{\tilde{t}_2}$ is indeed numerically negligible. We thus fix these values for concreteness to $m_{\tilde{q}} = 10$ TeV and $m_{\tilde{t}_2} = 10.01$ TeV in our computations. The effect of a variation of the remaining parameters $m_{\tilde{g}}$ and $\theta_{\tilde{t}}$ is in the percent range. In this context, the hierarchy between $m_{\tilde{g}}$ and $m_{\tilde{t}_1}$ is particularly relevant, since a light gluino facilitates an on-shell decay of the stop into a gluino and a top quark. We therefore keep both $m_{\tilde{t}_1}$ and $m_{\tilde{g}}$ as variable parameters in our results. Additionally, we encode the effect of varying the stop mixing angle $\theta_{\tilde{t}}$ within the range $\sin(2\theta_{\tilde{t}}) \in [-1, 1]$ in a relative parametric uncertainty $\delta_{\tilde{t},\text{param}\pm}$ with respect to the cross section evaluated with the default value of $\sin(2\theta_{\tilde{t}}) = 0.669$, corresponding to the CMSSM benchmark point 40.2.5 of [92]:

$$
\begin{aligned}
\delta_{\tilde{t},\text{param}+} &:= \frac{\max(\sigma_{\sin(2\theta_{\tilde{t}}) \in [-1,1]}) - \sigma_{\sin(2\theta_{\tilde{t}})=0.669}}{\sigma_{\sin(2\theta_{\tilde{t}})=0.669}}\,, \\
\delta_{\tilde{t},\text{param}-} &:= \frac{\min(\sigma_{\sin(2\theta_{\tilde{t}}) \in [-1,1]}) - \sigma_{\sin(2\theta_{\tilde{t}})=0.669}}{\sigma_{\sin(2\theta_{\tilde{t}})=0.669}}\,.
\end{aligned}
\tag{3.14}
$$

**Total theoretical uncertainty**   The total theoretical relative uncertainty $\delta_{\text{tot}\pm}$ on the calculated cross section is then given by all individual uncertainties as discussed above, i.e. $\delta_{\mu\pm,0}$ in Eq. (3.10) and $\delta_{\text{PDF}+\alpha_{\text{s}}}$ in Eq. (3.13), added in quadrature,[6]

$$\delta_{\text{tot}+} := \sqrt{(\delta_{\mu+,0})^2 + (\delta_{\text{PDF}+\alpha_{\text{s}}})^2}\,, \qquad \delta_{\text{tot}-} := \sqrt{(\delta_{\mu-,0})^2 + (\delta_{\text{PDF}+\alpha_{\text{s}}})^2}\,, \tag{3.15}$$

---

[5]By 'light-flavoured' squarks, we mean the superpartners of the light quark flavours.

[6]A more conservative approach would rely on adding the uncertainties linearly. However, for high enough masses the PDF error vastly dominates, and the difference between adding PDF and scale errors linearly or quadratically is minimal.

including also $\delta_{\tilde{t},\text{param}\pm}$ in Eq. (3.14) in the case of stop production:

$$\delta_{\text{tot}+} := \sqrt{(\delta_{\mu+,0})^2 + (\delta_{\text{PDF}+\alpha_\text{s}})^2 + (\delta_{\tilde{t},\text{param}+})^2}\,,$$
$$\delta_{\text{tot}-} := \sqrt{(\delta_{\mu-,0})^2 + (\delta_{\text{PDF}+\alpha_\text{s}})^2 + (\delta_{\tilde{t},\text{param}-})^2}\,. \tag{3.16}$$

We can then define an upper ($U$) and lower ($L$) limit of the cross section prediction as:

$$U := \sigma_{\text{central}}\left(1 + \delta_{\text{tot}+}\right), \qquad L := \sigma_{\text{central}}\left(1 - \delta_{\text{tot}-}\right), \tag{3.17}$$

where $\sigma_{\text{central}}$ denotes the cross section calculated with central values for the scale, the PDF member, the $\alpha_\text{s}$ value, and, if applicable, the stop mixing angle. We now define in terms of $U$ and $L$ the average cross section $\sigma_{\text{avg}}$ as well as the combined symmetric theoretical uncertainty $\delta_{\text{sym}}$:

$$\sigma_{\text{avg}} := \frac{U+L}{2}\,, \qquad \delta_{\text{sym}} := \frac{U-L}{2\sigma_{\text{avg}}} = \frac{U-L}{U+L}\,. \tag{3.18}$$

The presented way of treating and combining the uncertainties, now with the modern PDF4LHC21 set, is in agreement with the previous approach taken in [66] for the calculation of squark and gluino cross sections at NLO+NLL accuracy.

## 3.4 Non-degenerate squark masses

As mentioned previously, while calculating the $\text{NNLO}_{\text{Approx}}+\text{NNLL}$ predictions according to Eq. (3.7), we assume an 8- or 10-fold degeneracy among the light-flavoured squark masses, so that the cross section only depends on one squark mass parameter $m_{\tilde{q}}$. To compute the cross section predictions for an MSSM parameter point with non-degenerate squark masses, the parameter $m_{\tilde{q}}$ should then be chosen as the average value of all light-flavoured squark masses other than $\tilde{t}_1$, $\tilde{t}_2$ (and $\tilde{b}_1$, $\tilde{b}_2$, if appropriate).

In case cross section predictions for non-degenerate squark masses are needed, we propose as a prescription to rescale the NNLL-FAST cross section obtained by Eq. (3.7) by the factor

$$R_{\text{non-deg.}} := \frac{\sigma_{h_1 h_2 \to kl}^{\text{LO, non-deg.}}(m_{\tilde{u}_L}, m_{\tilde{u}_R}, m_{\tilde{d}_L}, m_{\tilde{d}_R}, \ldots)}{\sigma_{h_1 h_2 \to kl}^{\text{LO, deg.}}(m_{\tilde{q}})}\,, \tag{3.19}$$

where $\sigma_{h_1 h_2 \to kl}^{\text{LO, non-deg.}}$ is the LO cross section for the squark and gluino production process with all squark masses considered non-degenerate, while $\sigma_{h_1 h_2 \to kl}^{\text{LO, deg.}}$ is the corresponding LO cross section with degenerate squark masses, which can both be obtained from e.g. PROSPINO. Then, the approximation of the total cross section with non-degenerate

squark masses is given as:

$$\sigma_{h_1 h_2 \to kl}^{\text{NNLL-FAST, non-deg.}} = R_{\text{non-deg.}} \times \sigma_{h_1 h_2 \to kl}^{\text{NNLL-FAST}} . \qquad (3.20)$$

Note that this is the same procedure as implemented in the PROSPINO 2 code to compute approximate NLO-QCD predictions for non-degenerate squarks.

The quality of this approximation was studied in the past for selected pre-Run 2 benchmarks points (e.g. in [40, 42, 43]). When a sum of the cross sections over different flavour and chirality combinations was considered, the studies showed only negligible differences between NLO-QCD $K$-factors, i.e. ratios of the NLO over the LO cross section, calculated with non-degenerate and degenerate squark masses. Based on the observed behaviour of the NLO cross section, as well as the proportionality of $\Delta\text{NNLO}_{\text{Approx}}$ to the LO cross section, we expect similarly negligible effects from non-degenerate squark masses for $\sigma_{h_1 h_2 \to kl}^{\text{NNLO}_{\text{Approx}}}$. In addition, the bound-state contributions, as well as the threshold-resummed NNLL corrections are flavour-blind. Therefore the same conclusion must hold, i.e. accounting for squark mass degeneracy by rescaling with the ratio of Eq. (3.19) as done in Eq. (3.20) provides a very good approximation.

# 4 NNLL-fast

The cross sections for squark and gluino hadroproduction at $\text{NNLO}_{\text{Approx}}+\text{NNLL}$ accuracy, evaluated according to Eqs. (3.7) and (3.1) with the PDF4LHC21 set at the LHC Run 3 collision energy of $\sqrt{S} = 13.6$ TeV, together with all the associated theoretical uncertainties, are provided in the version 2.0 of the publicly available code NNLL-FAST. The package is a successor to the NLL-FAST project [60–67].

The NLO SUSY-QCD cross section is computed using the PROSPINO 2 code [85]. The remaining terms in Eqs. (3.7) and (3.1), i.e. the threshold-enhanced approximated NNLO corrections $\Delta\sigma^{\text{NNLO}_{\text{Approx}}}$, the bound-state contributions $\sigma^{\text{BS}}$, as well as the soft-gluon and Coulomb resummed contributions beyond NNLO accuracy are calculated and cross-checked with two in-house codes, for which we find very good numerical agreement. NNLL-FAST implements all processes described in Sec. 2. In addition to these processes, we also provide predictions for gluino-pair production in the limit of decoupled, i.e. very heavy, squarks, and squark-antisquark production in the limit of decoupled gluinos. The uncertainties are computed according to Eqs. (3.9) and (3.13) for the variation of the renormalisation and factorisation scales and the combined PDF+$\alpha_{\text{s}}$ uncertainty, respectively, and, in the case of stop-antistop production, according to Eq. (3.14) for

the variation of the remaining SUSY parameters. The results in NNLL-FAST for the total cross sections and the associated uncertainties are given in form of numerical grids, together with a fast interpolation code. The mass ranges of $m_{\tilde{q}/\tilde{t}_1}$ and $m_{\tilde{g}}$ for the grids are the following:[7]

- $\tilde{g}\tilde{g}$, $\tilde{q}\tilde{q}^*$, $\tilde{q}\tilde{g}$, and $\tilde{q}\tilde{q}$ production:

$$m_{\tilde{q}} \in [500, 3000]\ \text{GeV}, \qquad m_{\tilde{g}} \in [500, 3000]\ \text{GeV}, \tag{4.1}$$

- $\tilde{g}\tilde{g}$ production with decoupled squarks ($m_{\tilde{q}}$ chosen very heavy):

$$m_{\tilde{g}} \in [500, 3000]\ \text{GeV}, \tag{4.2}$$

- $\tilde{q}\tilde{q}^*$ production with decoupled gluinos ($m_{\tilde{g}}$ chosen very heavy):

$$m_{\tilde{q}} \in [500, 3000]\ \text{GeV}, \tag{4.3}$$

- $\tilde{t}_1\tilde{t}_1^*$ production:

$$m_{\tilde{t}_1} \in [100, 3000]\ \text{GeV}, \qquad m_{\tilde{g}} \in [500, 5000]\ \text{GeV}. \tag{4.4}$$

Compared to the previous version 1.1 of NNLL-FAST, the technical work on the version 2.0 update consisted of generating the new grids containing the NLO-QCD and the threshold-resummation-improved $\text{NNLO}_{\text{Approx}}+\text{NNLL}$ cross sections together with the associated uncertainties, evaluated with the PDF4LHC21 set at $\sqrt{S} = 13.6$ TeV. Additionally, checks regarding the interpolation quality for cross sections in between grid points were performed, to make sure that the interpolated results are in agreement with those obtained from a direct computation. We found up to 2% discrepancies in between grid points for processes other than $\tilde{t}_1\tilde{t}_1^*$ production, where the interpolation error could reach 5%. These maximal values were encountered mostly at the extreme edges of the grids (i.e. very small or very large masses). For $\tilde{g}\tilde{g}$ production, interpolation errors of 1–2% are observed also for some intermediate points. Other than these singular cases, the interpolation accuracy was found to always be better than 1%.

---

[7]Extended mass ranges are available on request. For specific processes, tabulated cross sections for mass values outside of the mentioned ranges are available on the TWiki page of the LHC SUSY Cross Section Working Group https://twiki.cern.ch/twiki/bin/view/LHCPhysics/SUSYCrossSections.

| Label | Production process |
|:-----:|:------------------:|
| gg | $\tilde{g}\tilde{g}$ |
| sb | $\tilde{q}\tilde{q}^*$ |
| sg | $\tilde{q}\tilde{g}$ |
| ss | $\tilde{q}\tilde{q}$ |
| st | $\tilde{t}_1\tilde{t}_1^*$ |
| gdcpl | $\tilde{g}\tilde{g}$ with decoupled squarks |
| sdcpl | $\tilde{q}\tilde{q}^*$ with decoupled gluinos |

**Table 1:** Abbreviations for the production processes of squark, gluino, and stop production when running the NNLL-FAST code. For specific running signatures, in particular for the `st`, `gdcpl`, and `sdcpl` processes, see the corresponding text below.

## 4.1 Running of the code

The NNLL-FAST 2.0 code and its previous versions are made available under the following link:

https://www.uni-muenster.de/Physik.TP/~akule_01/nnllfast

After downloading the NNLL-FAST 2.0 package and unpacking it, the interpolation code written in Fortran can be compiled within the `nnllfast-2.0/` directory by typing in a terminal the following command, assuming the GNU Fortran compiler of the GNU Compiler Collection to be used:

```
gfortran nnllfast-2.0.f -o name_of_the_executable
```

The name of the executable `name_of_the_executable` can be chosen freely. The executable can then be called to obtain cross section results including the associated theoretical uncertainties:

```
./name_of_the_executable <process> <squark_mass> <gluino_mass>
```

where `<process>` is one of the labels listed in Table 1, and `<squark_mass>` as well as `<gluino_mass>` correspond to the pair of values for $m_{\tilde{q}}$ as well as $m_{\tilde{g}}$ for which the cross section should be output. In the case of stop production, the second argument of the squark mass `<squark_mass>` is replaced by `<stop_mass>` corresponding to the light stop mass $m_{\tilde{t}_1}$:

```
./name_of_the_executable st <stop_mass> <gluino_mass>
```

In the case of gluino-pair production with decoupled squarks or squark-antisquark production with decoupled gluinos, the executable should be called as:

```
 ./name_of_the_executable gdcpl <gluino_mass>
```

with <gluino_mass> set to the value of choice for $m_{\tilde{g}}$, or

```
 ./name_of_the_executable sdcpl <squark_mass>
```

with <squark_mass> set to the required value of $m_{\tilde{q}}$, respectively.

An example output for the following command line

```
 ./name_of_the_executable sg 1700 2100
```

is:

```
# LHC @ 13.6 TeV, NNLO PDF4LHC21 (LHAPDF ID 93300)
# process: sg
# ms[GeV] mg[GeV]  NLO[pb]   NNLL+NNLO_app[pb] d_mu+[%] d_mu-[%] d_pdfas+[%] d_pdfas-[%] K_NNLL
-------------------------------------------------------------------------------------------
  1700.   2100.   0.733E-02   0.891E-02         3.69    -5.49     9.86       -9.86      1.21
```

Here, the first two columns denote the input values of $m_{\tilde{q}}$ and $m_{\tilde{g}}$, the third column corresponds to the fixed-order NLO-QCD cross section, and the fourth column corresponds to the NNLL-FAST cross section of Eq. (3.7) at $NNLO_{Approx}+NNLL$ accuracy including threshold-resummation corrections. Columns five to eight correspond to the upper and lower scale uncertainties $\delta_{\mu\pm}$ of Eq. (3.9) and the PDF+$\alpha_s$ uncertainty $\delta_{PDF+\alpha_s}$ of Eq. (3.13) given in percent. In the last column, we output the $K_{NNLL}$ factor given as the ratio of the NNLL-FAST cross section over the NLO-QCD result,

$$K_{NNLL} := \frac{\sigma^{NNLL\text{-}FAST}}{\sigma^{NLO}} , \tag{4.5}$$

which denotes the size of the threshold-enhanced corrections beyond NLO.

# 5 Numerical results

In this section, we present numerical results based on $NNLO_{Approx}+NNLL$ calculations which can be obtained with the NNLL-FAST package. Note that here and in the following, we use for the accuracy the labels "$NNLO_{Approx}+NNLL$" and "NNLL-FAST" interchangeably, and we always mean our best accuracy by including all terms according to Eqs. (3.7). Unless otherwise stated, the results are obtained with the PDF4LHC21 Hessian set (LHAPDF ID 93300), accessed through the LHAPDF 6 library [93], and at a centre-of-mass energy of $\sqrt{S} = 13.6$ TeV, using a common renormalisation and factorisation scale $\mu$ which has been set to the central scale choice of the average mass of the produced particles, $\mu = \mu_0 = m_{av}$, as discussed above. We note that the PDF4LHC

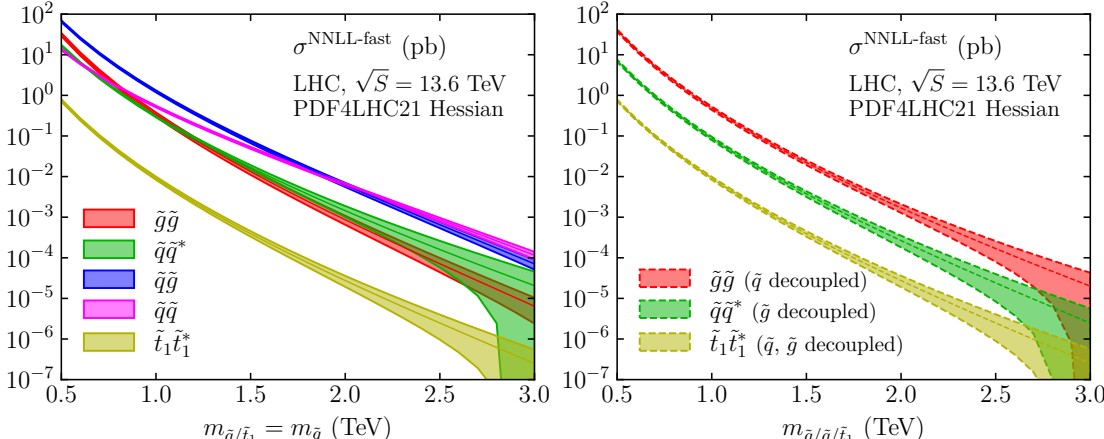

**Figure 2:** Total inclusive cross sections for squark, gluino, and stop production at the LHC at $\sqrt{S} = 13.6$ TeV. Left: equal $m_{\tilde{q}} = m_{\tilde{g}}$ (or $m_{\tilde{t}_1} = m_{\tilde{g}}$ for $\tilde{t}_1\tilde{t}_1^*$, respectively). Right: decoupled scenarios. The bands denote the uncertainty envelope around the central cross section values as discussed in Eq. (3.17). As we consider all light-flavour squarks to be degenerate, the letter $\tilde{q}$ implicitly stands for the sum over all squark flavours, and the production cross sections consist of the sum of each possible flavour combination.

collaboration offers their recent PDF4LHC21 sets only at NNLO accuracy, so that in the following discussion, both the NLO as well as the $\mathrm{NNLO_{Approx}{+}NNLL}$ cross sections are evaluated with the same NNLO PDFs.

## 5.1 Predictions for $\sqrt{S} = 13.6$ TeV

The dependence of the total cross section for all processes of interest on the mass of the produced sparticle in the final state is shown in Fig. 2. The central cross section values as well as the uncertainty bands are computed according to Eq. (3.17). The width of the bands denotes the total theoretical uncertainty, calculated as described in section 3.3. The left plot of Fig. 2 displays the cross sections under the assumption of equal squark and gluino masses, $m_{\tilde{q}} = m_{\tilde{g}}$ (or $m_{\tilde{t}_1} = m_{\tilde{g}}$ for stop-antistop production), while the right plot presents the case of the decoupled scenarios, i.e. where either the squarks (in the case of $\tilde{g}\tilde{g}$), or the gluinos ($\tilde{q}\tilde{q}^*$), or both ($\tilde{t}_1\tilde{t}_1^*$) are assumed very heavy[8]. In the cases of $\tilde{q}\tilde{q}^*$, $\tilde{q}\tilde{g}$, and $\tilde{q}\tilde{q}$, the ten light-flavour squarks are considered as degenerate, i.e. they all have the same mass $m_{\tilde{q}}$, and the production cross section shown corresponds to a sum over all degenerate final states.

---

[8]We checked that for $\tilde{t}_1\tilde{t}_1^*$, while the squarks are always chosen to be decoupled at $m_{\tilde{q}} = 10$ TeV due to their negligible impact on the cross section, a value of $m_{\tilde{g}} = 5$ TeV is sufficiently high to consider stop production in the decoupling regime, as the cross section remains constant even for higher gluino masses such as $m_{\tilde{g}} = 10$ TeV.

For equal squark and gluino masses, it can be seen in the left plot of Fig. 2 that while for low masses, the processes where one or two gluinos are being produced dominate over the other processes, the cross section of squark-pair production drops less rapidly and becomes the dominant process for large squark and gluino masses. This effect is related to the parton luminosities: the process $\tilde{q}\tilde{q}$ can proceed via the collision of two valence quarks, while all other processes depend, at LO, through their initial states on antiquark or gluon PDFs, which, at high masses and consequently high momentum fractions of the partons, are more strongly suppressed than the valence-quark PDFs. In the case of the decoupled scenarios, the right plot of Fig. 2 shows that, while the $\tilde{g}\tilde{g}$ cross section is of similar size for small masses as in the equal-mass case, it reaches larger values for large gluino masses as compared to the equal-mass case. The opposite behaviour can be seen for $\tilde{q}\tilde{q}^*$, where the cross section of the decoupled case always lies below the equal-mass case. In the case of $\tilde{t}_1\tilde{t}_1^*$, there is almost no difference between the equal-mass and decoupled cases due to the dependence on $m_{\tilde{q}}$ and $m_{\tilde{g}}$ arising only from higher orders.

There exists no decoupling limit for $\tilde{q}\tilde{g}$ as both squarks and gluinos appear in the final state, and the cross section thus tends towards zero for very heavy $m_{\tilde{q}}$ or $m_{\tilde{g}}$. Similarly, the $\tilde{q}\tilde{q}$ cross section becomes zero for decoupled $\tilde{g}$, as can be seen e.g. from the tree-level diagrams of Fig. 1 (d), where the gluino appears as a virtual particle in all diagrams in the $t$- or $u$-channel, respectively, and the amplitudes are thus heavily suppressed for very large $m_{\tilde{g}}$.

We note that for $\tilde{q}\tilde{q}^*$ and $\tilde{t}\tilde{t}^*$ in the equal-mass case as well as for all processes in the decoupled case, the uncertainty band towards large mass values becomes very large and the error surpasses 100%, causing the lower end of the band to extend towards very small values in the plots with a logarithmic axis.

### 5.1.1 PDF+$\alpha_\mathrm{s}$ and scale uncertanties

In the following, we discuss the sources of theoretical uncertainties for the processes of squark, gluino, and stop production, as discussed in Sec. 3.3. In Figs. 3 (for the equal-mass case) and 4 (for the decoupled scenarios), we show the relative sizes of the PDF+$\alpha_\mathrm{s}$ uncertainties as well as the uncertainty related to the variation of the common renormalisation and factorisation scale $\mu$.

We find that going from the best fixed-order prediction NLO to NNLO$_\mathrm{Approx}$+NNLL, the theoretical scale uncertainty is reduced significantly for all processes and is almost constant with respect to the masses of the produced particles. The strongest reduction is found for $\tilde{q}\tilde{q}^*$ in the equal-mass case, and for $\tilde{t}_1\tilde{t}_1^*$ in the decoupled scenario. For the shown mass ranges, the scale uncertainties are of the order of or below 10% for all

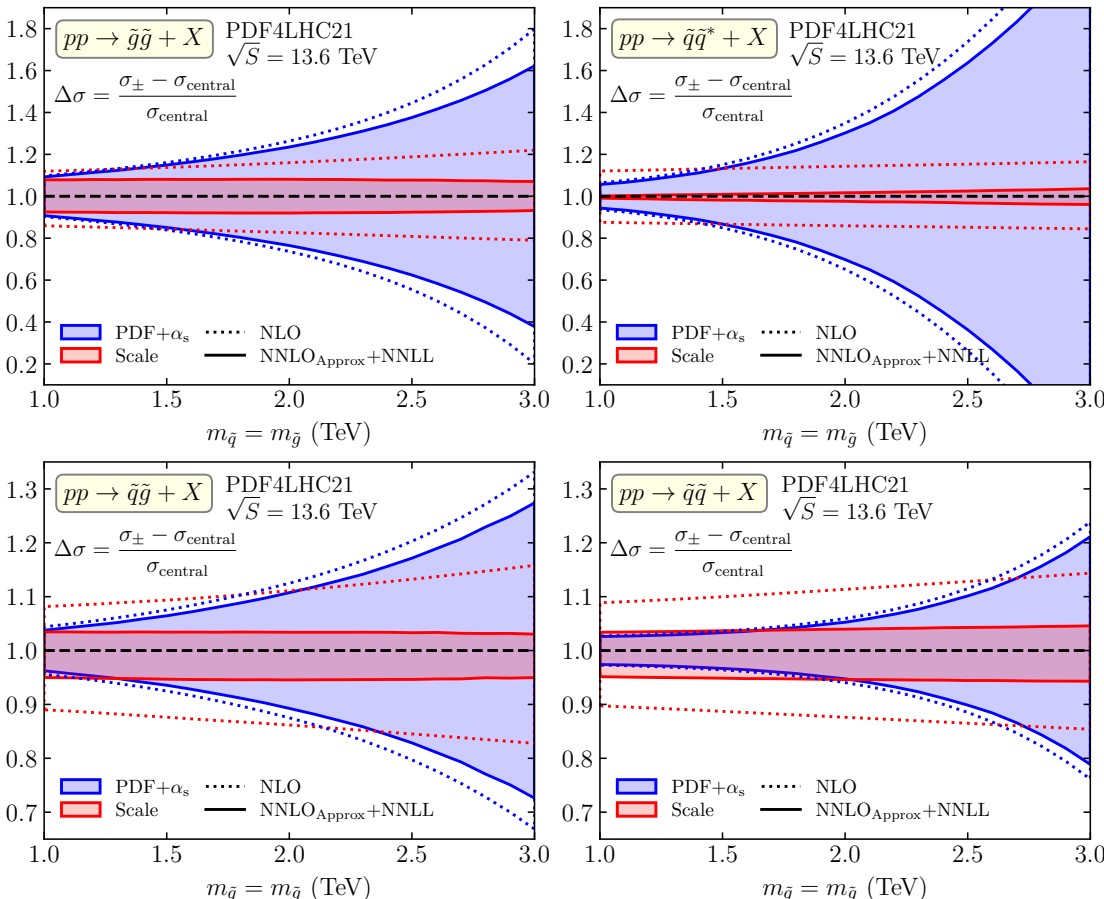

**Figure 3:** Theoretical uncertainties for $\tilde{g}\tilde{g}$, $\tilde{q}\tilde{q}^*$, $\tilde{q}\tilde{g}$, and $\tilde{q}\tilde{q}$ production in the case of equal $m_{\tilde{q}} = m_{\tilde{g}}$ at NNLO$_{\text{Approx}}$+NNLL with a centre-of-mass energy of $\sqrt{S} = 13.6$ TeV and the PDF4LHC21 set, as provided by NNLL-FAST 2.0. For the scale uncertainties, $\sigma_{\pm} := \sigma_{\text{central}} \left(1 \pm \delta_{\mu\pm,0}\right)$ with $\delta_{\mu\pm,0}$ from Eq. (3.10), and for the PDF+$\alpha_{\text{s}}$ uncertainties, $\sigma_{\pm} := \sigma_{\text{central}} \left(1 \pm \delta_{\text{PDF}+\alpha_{\text{s}}}\right)$ with $\delta_{\text{PDF}+\alpha_{\text{s}}}$ from Eq. (3.13).

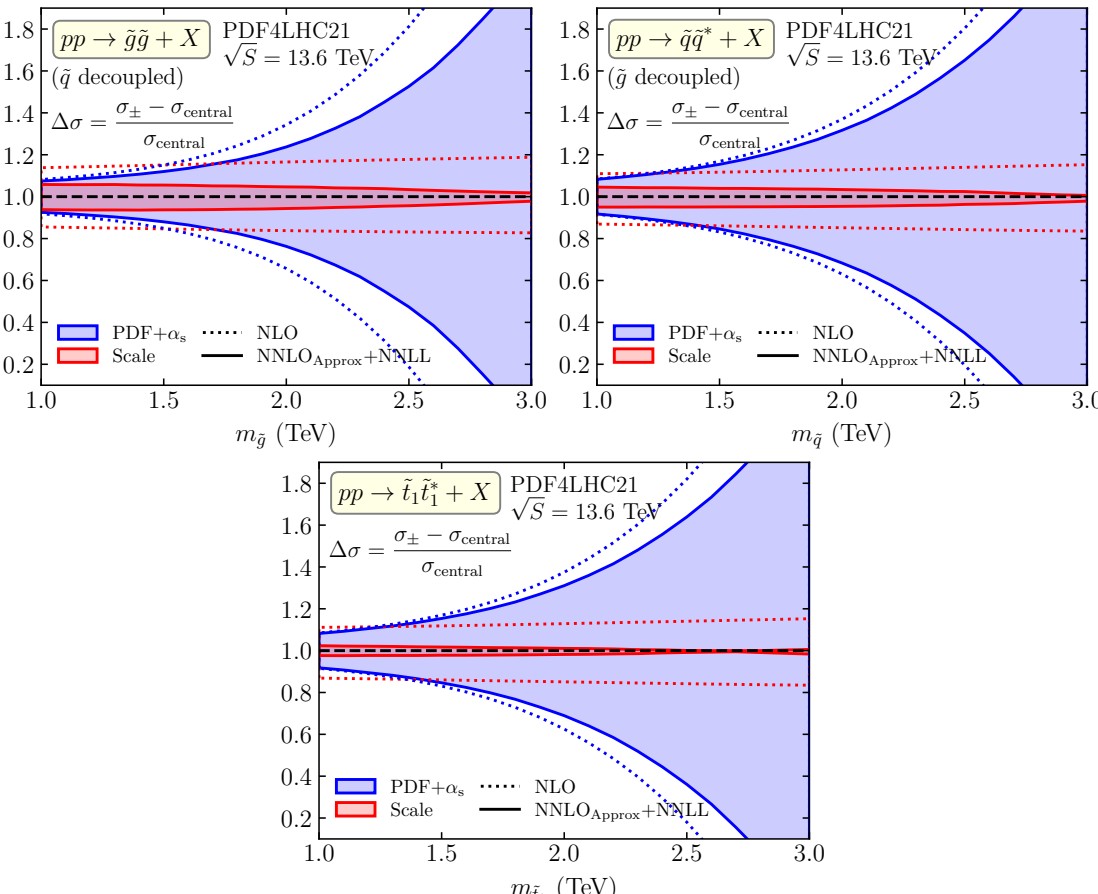

**Figure 4:** Theoretical uncertainties for $\tilde{g}\tilde{g}$, $\tilde{q}\tilde{q}^*$, and $\tilde{t}_1\tilde{t}_1^*$ production in the decoupled scenarios, where all sparticles other than the produced ones are assumed very heavy, with the otherwise same set-up as for Fig. 3.

processes.

In contrast, the uncertainty due to the parametrisation of the PDFs as well as a variation of the value of $\alpha_{\mathrm{s}}$ is not affected by the increase in accuracy to the same degree as the scale uncertainty. As noted before, both the NLO as well as $\mathrm{NNLO_{Approx}+NNLL}$ cross sections are computed with the same PDFs at NNLO accuracy, so we do not expect a significant improvement of the PDF uncertainties. We nonetheless see a slight decrease of the PDF+$\alpha_{\mathrm{s}}$ uncertainty, in particular towards higher masses where the uncertainty becomes large, which is related to cancellations of higher-order terms between the PDF evolution and the threshold effects beyond NLO. After including resummation corrections, the PDF+$\alpha_{\mathrm{s}}$ uncertainty now constitutes the dominant source of uncertainty for all processes, with the exception of $\tilde{q}\tilde{q}$ where the scale uncertainty is of the same order or slighly above the PDF+$\alpha_{\mathrm{s}}$ uncertainty up to about $m_{\tilde{q}} = m_{\tilde{g}} = 2$ TeV. As mentioned already in the discussion of Fig. 2, for the processes of $\tilde{q}\tilde{q}^*$ and $\tilde{t}_1\tilde{t}_1^*$ as well as $\tilde{g}\tilde{g}$ in the decoupled case, the PDF+$\alpha_{\mathrm{s}}$ uncertainty grows above 100% at high masses, where the gluon initial states dominate, due to a lack of data to constrain in particular the gluon and sea quark luminosities at high scales.

## 5.2 Comparison to previous results for $\sqrt{S} = 13$ TeV

We move on to a discussion of the differences between our current results for Run 3 as presented in this paper and the previous predictions from 2016 for Run 2 at $\sqrt{S} = 13$ TeV [77] and computed with the PDF4LHC15 set [94]. The comparison is shown in Fig. 5 as a ratio of the central $\mathrm{NNLO_{Approx}+NNLL}$ cross sections obtained by the recent NNLL-FAST 2.0 to the previous NNLL-FAST 1.1 results for each process. We probe the parameter space by presenting the dependence on the mass of the produced sparticle for a selected range of values of the other mass parameter ($m_{\tilde{g}}$ in the case of squark production, $m_{\tilde{q}}$ for the gluino production). The decoupled cases, as discussed before, are denoted in the plots by the dashed lines, wherever applicable. For all processes, the ratio is growing with increasing masses of the produced particles. With the exception of $\tilde{q}\tilde{g}$ production, the dependence of the ratio on the other mass parameter is relatively small, and in most cases begins to be visible only for very large masses of the produced particles. The modification of the cross section, illustrated by the ratio, ranges from a few tens of percent to a factor of a few, with the highest factor of about 4.5 observed for $\tilde{q}\tilde{q}^*$, and the smallest of about 1.6 for $\tilde{q}\tilde{q}$ production at high masses.

It is interesting to study where the effect is coming from. To this end, in Figs. 6 and 7, we show separately the impact of the change of the PDF set and of the increase in the collision energy, respectively. While the ratio for changing the PDF set is strongly influ-

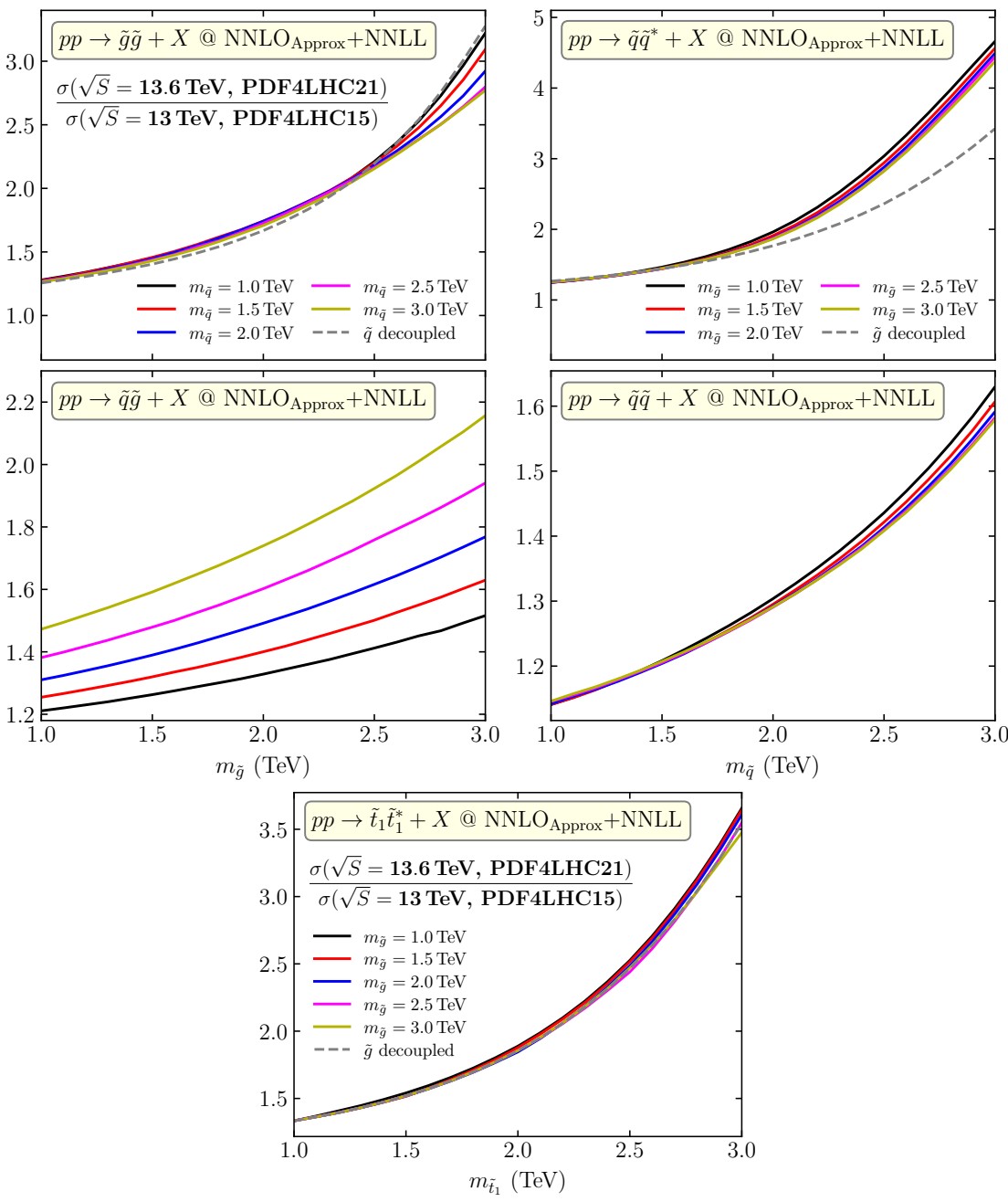

**Figure 5:** Comparison between the central $\text{NNLO}_{\text{Approx}}+\text{NNLL}$ cross sections for squark, gluino, and stop production as provided by NNLL-FAST 2.0 (for $\sqrt{S} = 13.6$ TeV and with PDF4LHC21) and the previous version NNLL-FAST 1.1 (for $\sqrt{S} = 13$ TeV and with PDF4LHC15), presented as the ratio of the two. The ratios are shown depending on the mass of the produced sparticle for a range of values of the other mass parameter.

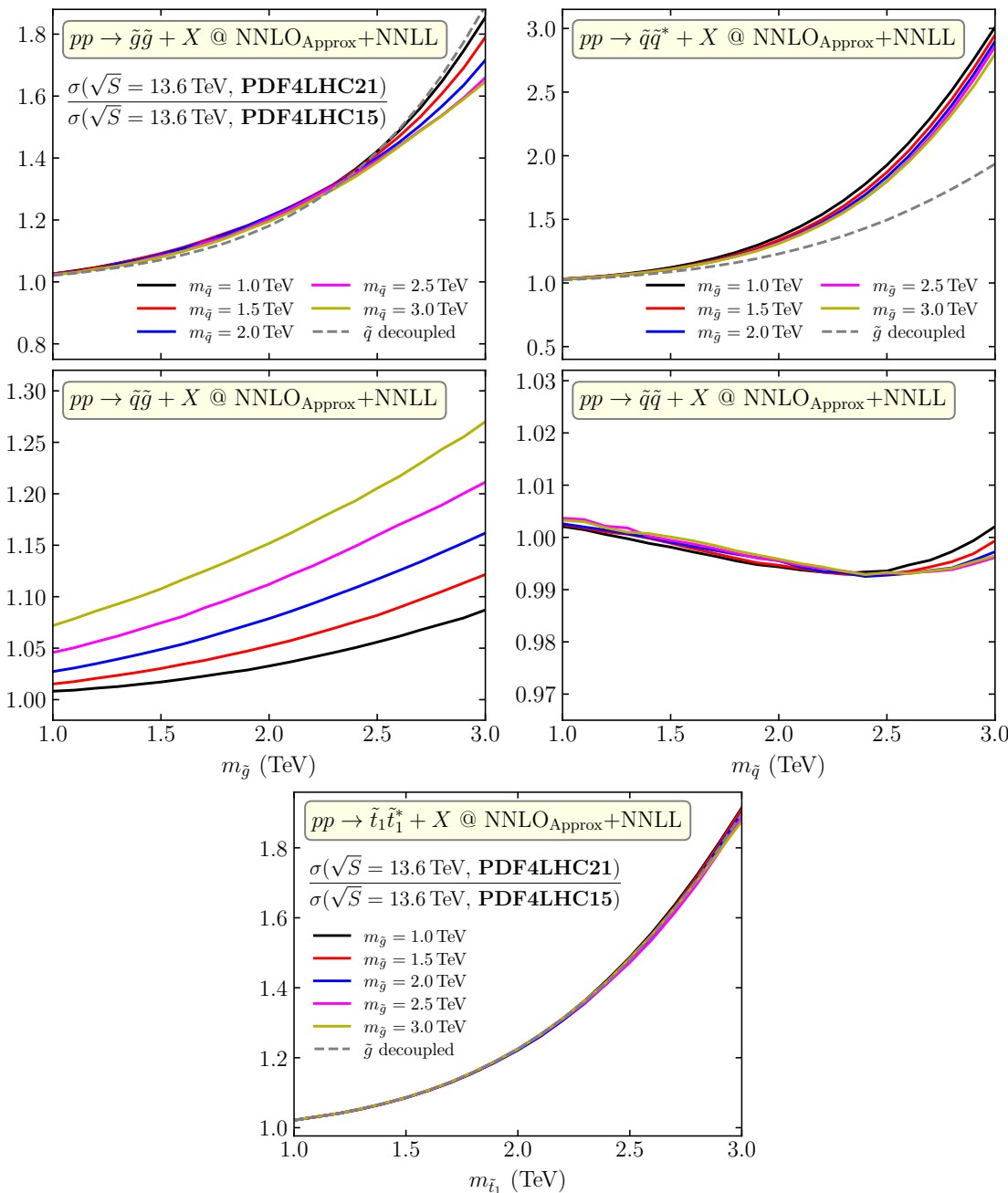

**Figure 6:** Same as Fig. 5, but with a fixed centre-of-mass energy ($\sqrt{S} = 13.6$ TeV) and different PDFs used in the numerator (PDF4LHC21) and denominator (PDF4LHC15).

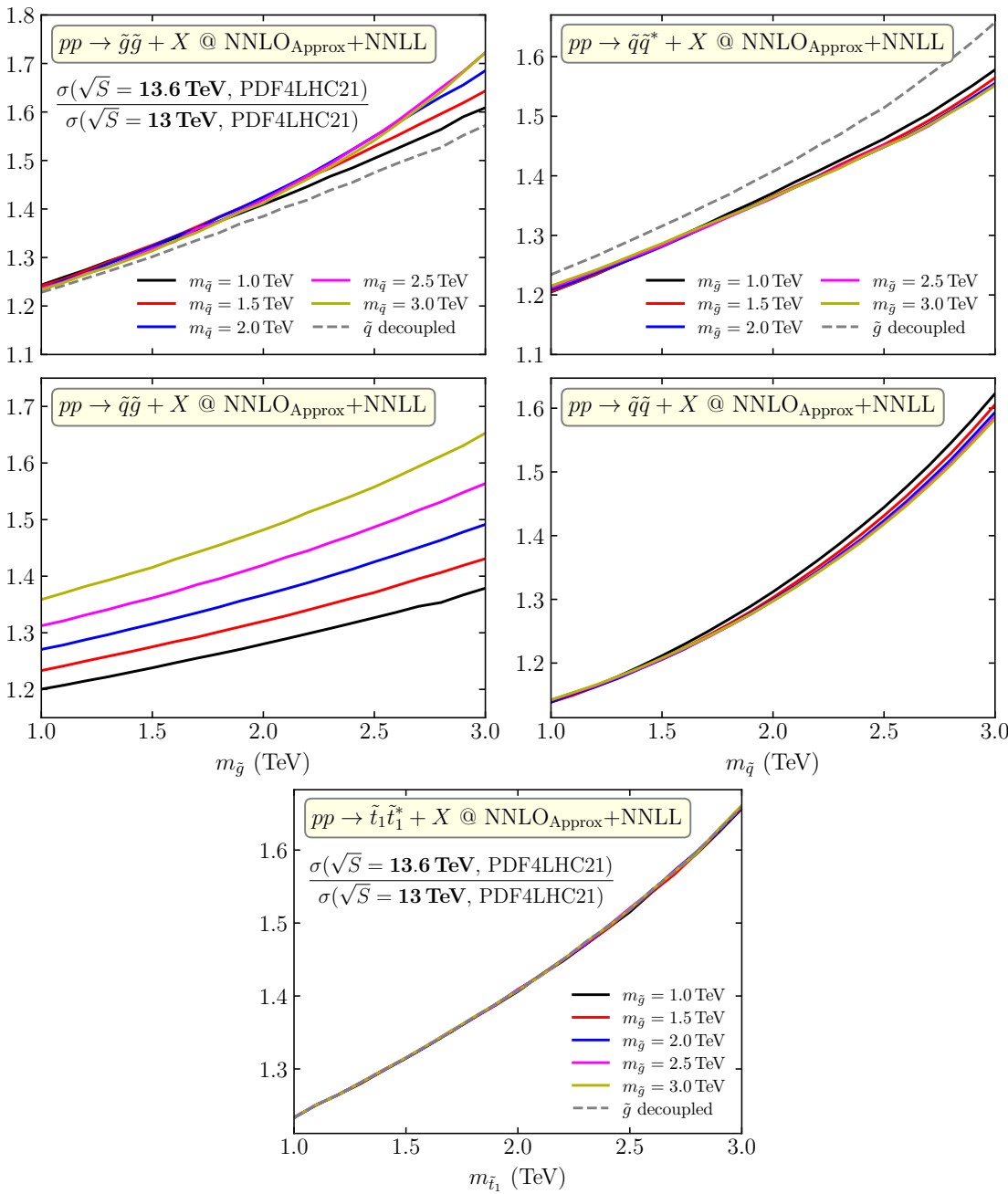

**Figure 7:** Same as Fig. 5, but with a fixed PDF set (PDF4LHC21) and different centre-of-mass energies used in the numerator ($\sqrt{S} = 13.6$ TeV) and the denominator ($\sqrt{S} = 13$ TeV).

enced by the different processes depending on different PDF luminosities, the increase due to changing the centre-of-mass energy is approximately the same for all processes and accounts for up to about 60–70% at high masses. In particular, we find that the increase in the $\tilde{q}\tilde{q}$ cross section is solely driven by calculating it at a higher centre-of-mass energy, whereas for all other processes the increase can be traced back to both higher energy collisions and the different set of PDFs. This is in agreement with the $\tilde{q}\tilde{q}$ production taking place only in the $qq$ channel at LO and the $qq$ luminosities being very similar for both PDF sets [88].

An analogous comparison of the NNLO$_{\text{Approx}}$+NNLL results obtained for a centre-of-mass energy of $\sqrt{S} = 14$ TeV with the PDF4LHC21 set and the results for Run 2 provided by NNLL-FAST 1.1 is presented in Fig. 8 and shows higher values of the corresponding ratios, but a qualitatively similar dependence on the $\tilde{q}$ and $\tilde{g}$ masses as in the case of $\sqrt{S} = 13.6$ TeV in Fig. 5.

## 5.3 Comparison between $K$-factors for $\sqrt{S} = \{13, 13.6, 14\}$ TeV

Another important information in the context of NNLO$_{\text{Approx}}$+NNLL calculations is the size of the NNLL corrections, as compared to the best fixed-order predictions, i.e. NLO. In Fig. 9, we thus show the corresponding $K$-factor as defined in Eq. (4.5), i.e. the ratio between the central NNLO$_{\text{Approx}}$+NNLL cross sections to the NLO results, for four different set-ups:

- for $\sqrt{S} = 13$ TeV with the PDF4LHC15 set, as provided by NNLL-FAST 1.1,

- for $\sqrt{S} = 13$ TeV with the PDF4LHC21 set,

- for $\sqrt{S} = 13.6$ TeV with the PDF4LHC21 set, as provided by NNLL-FAST 2.0,

- and for $\sqrt{S} = 14$ TeV with the PDF4LHC21 set.

Since the PDF4LHC21 set contains only the NNLO PDFs, we use them to also calculate the NLO cross sections entering the $K$-factors, contrary to the case of the PDF4LHC15 set where we use the NLO PDFs to compute the NLO cross sections in the $K$-factors. As expected, we observe that the relevance of resummed corrections diminishes slightly as the collision energy grows bigger. This can be deduced by comparing the $K$-factors for $\sqrt{S} = 13.6$ TeV and $\sqrt{S} = 14$ TeV, which are calculated with the same set of PDFs. Comparing the two $K$-factors obtained for $\sqrt{S} = 13$ TeV with the PDF4LHC15 and PDF4LHC21 sets shows however, that also the results for the $K$-factors are influenced by the PDFs, and that in a much stronger way than by the value of the collision energy. It

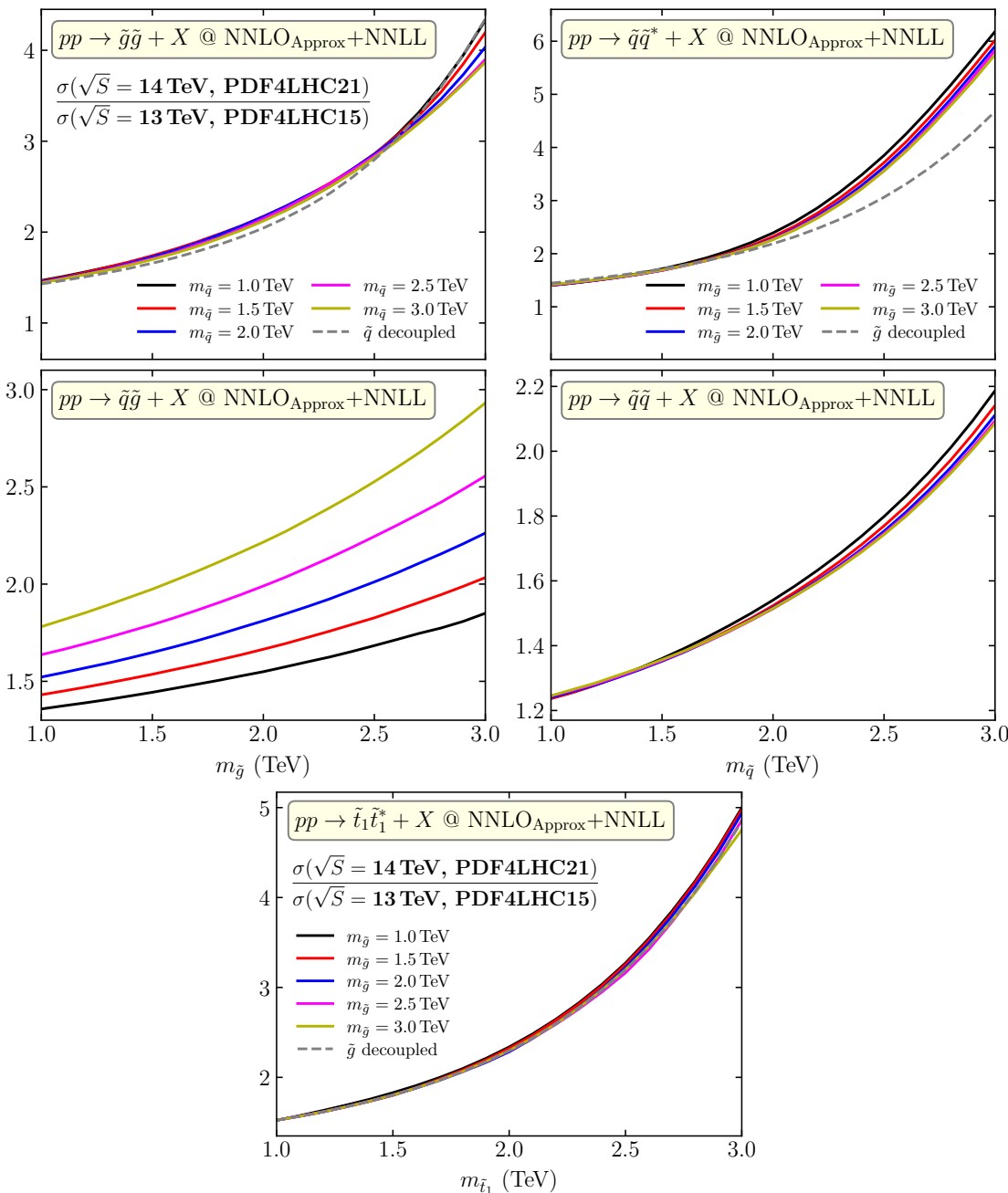

**Figure 8:** Same as Fig. 5, but with an increased centre-of-mass energy of $\sqrt{S} = 14$ TeV in the numerator.

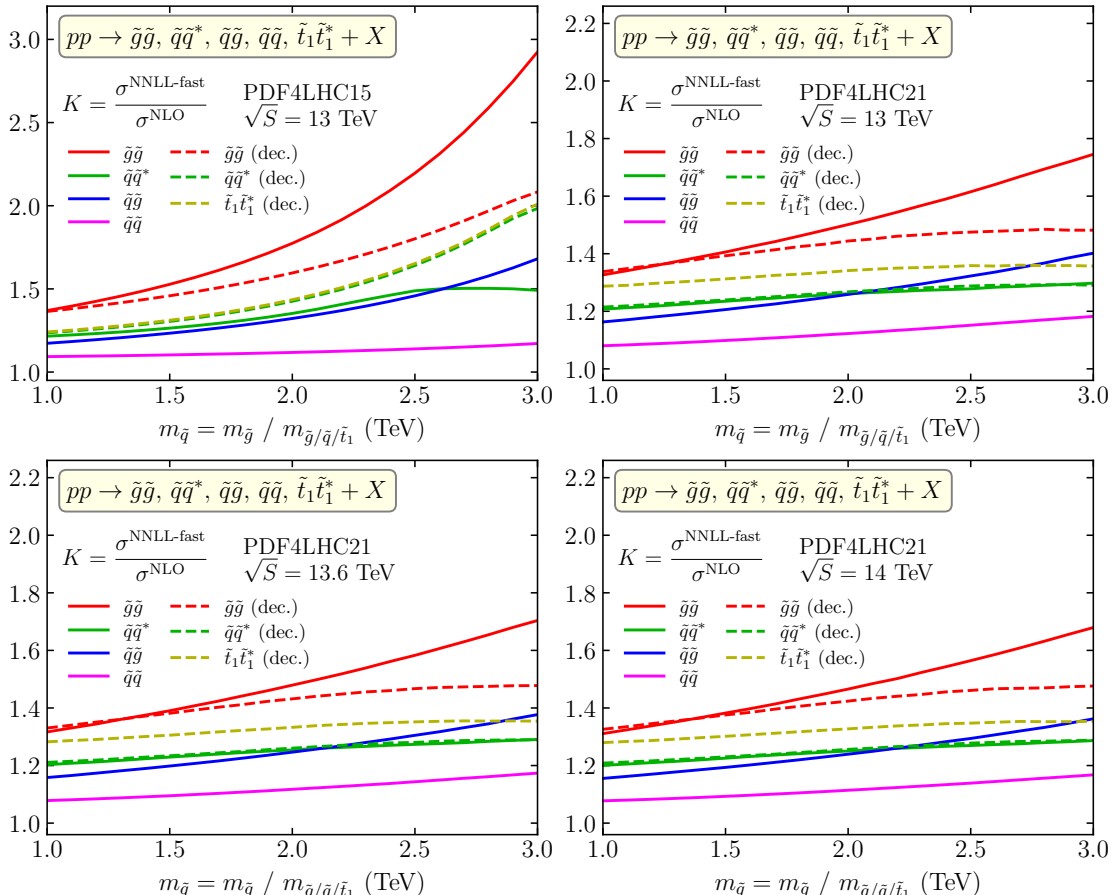

**Figure 9:** $K$-factors for all processes of squark and gluino production as well as stop production in the decoupled scenario. Top: $\sqrt{S} = 13$ TeV with PDF4LHC15 (left) and PDF4LHC21 sets (right). Bottom: $\sqrt{S} = 13.6$ TeV (left) and $\sqrt{S} = 14$ TeV (right), both with the PDF4LHC21 set.

should be noted that while all $K$-factors computed with the same NNLO PDF4LHC21 set behave similarly for different values of $\sqrt{S}$, the qualitative differences for the $K$-factors as provided by NNLL-FAST 1.1 are mainly due to NNLO PDFs of PDF4LHC15 being used for the $\mathrm{NNLO_{Approx}}$+NNLL cross sections in the numerator, and NLO PDFs of PDF4LHC15 being used for the NLO cross sections in the denominator.

For all discussed processes, the corrections due to threshold resummation are always positive in the shown mass ranges. The largest $K$-factors occur for $\tilde{g}\tilde{g}$ production, where in the equal-mass case at $\sqrt{S} = 13.6$ TeV, factors of up to 1.7 can be reached for $m_{\tilde{q}} = m_{\tilde{g}} = 3$ TeV. The corresponding $K$-factors for the decoupled scenario of $\tilde{g}\tilde{g}$ are a bit smaller and reach only a factor of about 1.5 for $\sqrt{S} = 13.6$ TeV. The lowest $K$-factors for all set-ups occur for $\tilde{q}\tilde{q}$ production where the effects due to threshold resummation are similar for all energies and increase the NLO cross section only by approximately 10–20%.

## 6  Conclusions and outlook

In this work, we report on the update of the predictions for coloured sparticle production at the LHC Run 3 for a collision energy of $\sqrt{S} = 13.6$ TeV using the updated PDF4LHC21 set. The predictions for the total cross sections of the processes of gluino-pair, squark-antisquark, squark-gluino, squark-pair as well as stop-antistop production were computed at $\mathrm{NNLO_{Approx}}$+NNLL accuracy including corrections from the threshold resummation of soft and Coulomb gluons as well as bound-state corrections. To date, the results constitute the state-of-the art theoretical predictions for these types of processes and are used by the experimental ATLAS and CMS collaborations for their analyses of squark and gluino searches. We furthermore describe the update to version 2.0 of the code package NNLL-FAST which includes the new predictions as numerical grids. Theoretical uncertainties are, as usual for higher-precision results, found to be reduced significantly compared to the fixed-order calculation at NLO-QCD, with the uncertainty now being dominated by the PDF error for most of the mass ranges.

In comparison to the previous version 1.1 of NNLL-FAST, we found that the total cross sections increased uniformly by up to about 60–70% in the probed mass regions for all processes by changing the centre-of-mass energy from 13 TeV to 13.6 TeV. The update from PDF4LHC15, as used in NNLL-FAST 1.1, to the newer PDF4LHC21 set influences the region of heavy squark and gluino masses such that the total effect on the ratio between the NNLL-FAST 2.0 and 1.1 results can reach a factor of above 2, and for some processes even a factor of above 4. We tested that for a future centre-of-mass

energy of $\sqrt{S} = 14$ TeV, the increase as compared to NNLL-FAST 1.1 for $\sqrt{S} = 13$ TeV could reach even higher factors of up to 6, which could turn out to be relevant for future precise determinations of the mass exclusion limits for squarks and gluinos in case of null results from SUSY searches, or, should a signal for beyond the Standard Model physics compatible with SUSY be found, to study the properties of the new particles.

The updated predictions are made available in form of numerical grids together with an interpolation code on the website of the NNLL-FAST project:

$$\text{https://www.uni-muenster.de/Physik.TP/\textasciitilde akule\_01/nnllfast}$$

Furthermore, for several simplified scenarios, the total cross section numbers for coloured sparticle production together with their uncertainties, calculated according to the expressions for the average cross section and the combined symmetric uncertainty in Eq. (3.18), are available for $\sqrt{S} = 13.6$ TeV and previous collision energies, amongst other sparticle production processes, on the TWiki page of the LHC SUSY Cross Section Working Group:

$$\text{https://twiki.cern.ch/twiki/bin/view/LHCPhysics/SUSYCrossSections}$$

## Acknowledgements

We are very grateful to our collaborators who have contributed to the NLL-FAST and NNLL-FAST projects over the years: S. Brensing-Thewes, R. Heger, M. Mangano, S. Marzani, L. Motyka, I. Niessen, J. Rojo, S. Padhi, T. Plehn, X. Portell, D. Schwartländer, and V. Theeuwes. MK acknowledges support from the DFG under grant 396021762 - TRR 257: Particle Physics Phenomenology after the Higgs Discovery. The work of LMV was supported by the DFG Research Training Group "GRK 2149: Strong and Weak Interactions - from Hadrons to Dark Matter".

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
