# Peer review of "NNLL-fast 2.0: Coloured Sparticle Production at the LHC Run 3 with $\sqrt{S}$ = 13.6 TeV"

_SciPost Physics_

## Round 1 · Referee Report · Anonymous (Referee 1) · 2024-6-26

Strengths

1) The paper is written clearly and describe concisely the procedure of threshold resummation (including the basic formulae) .

2) The NNLL-fast 2.0 source code is provided through link to the repository.

3) The results, their uncertainties, their dependence on other SUSY parameters, on PDFs and collisions' centre of mass energy are discussed in detail.

Report

The authors present the update of the numerical code NNLL-fast v.2.0, where they have implemented approximate NNLO + NNLL calculations for the production cross section of pairs of supersymmetric coulored particles (squarks, gluinos).
The new version of the code has been employed in the calculation of updated theoretical predictions for the cross section of many SUSY processes for the LHC Run 3 centre of mass energy of 13.6 TeV, adopting the PDF4LHC21 PDF set.
The results include estimations of the main sources of theoretical uncertainty (scale, PDF, $\alpha_s$), and discuss the dependence of the cross sections on the relevant SUSY parameters (masses, mixing angles).

The paper is certainly of great interest for the particle physics community as it provides the necessary theory predictions for the interpretation of experimental data in SUSY searches.
It is also very timely because experimental data is currently being collected at the LHC, and will be analysed in the upcoming years.

The authors' methodology is solid, they explain the basic command for the usage of the NNLL-fast code, and present and discuss the results with clarity.

The article meets the criteria for publication in this journal.
I recommend its publication in this journal, once the attached comments are addressed.

Requested changes

1) At page 10, below eq. 3.6, the authors state that the first term in the expansion of the matching coefficient $\mathcal{C}^{(1)}_{ij \rightarrow kl, I}$ collects the higher-order virtual contributions. I would think that this term would contain only the first higher-order corrections, i.e. the NLO corrections. Since the Authors are calculating cross sections with fixed order precision at approximate NNLO, I would think that in their calculations the Authors have included also the N-independent terms from $\mathcal{C}^{(2)}_{ij \rightarrow kl, I}$. Am I missing something? Is the Coulomb term in front perhaps already containing part of higher-order corrections?

2) At page 10, in the last paragraph, the authors describe their matching procedure to avoid double counting. This is just an English request. In the sentence: "To avoid the double counting of terms ..., and subtract it." the last "it" is maybe a bit too much implicit and unclear what it refers to. I do understand what the authors want to say, but perhaps the authors could spell it more clearly.

3) At page 11, below Eq. 3.7, the authors explain the various terms contained in the Equation. I understand it is far from the scope of this specific paper, but I was wandering if perhaps the authors can say something more about the $\sigma^{BS}$ term. For instance, how large is it compared to LO cross section, or NLO corrections? Is it more relevant for heavier final states sparticles?

4) At page 11, in the following sentence, the authors comment that the approximate NNLO cross section differs from the expanded resummed cross section by terms of order (1/N). I have the feeling that this sentence would fit better few lines above (just above Eq. 3.7), where the authors comment on the matching between fixed-order and resummed cross sections. This is just a personal taste, however I would recommend to at least add a comment about the (1/N) terms being suppressed in the threshold limit $N\rightarrow +\infty$.

5) At page 12, in the "PDF + $\alpha_s$" sub-sub-section, in the very first sentence, the authors define the PDF4LHC21 set the combination of various PDF sets in a global fit. This is not correct. The various PDF sets, CT18, MSHT20, NNPDF3.1 are the results of a global fit. PDF4LHC21 is not a global fit, but rather it is a Monte Carlo combination of the other global fits. In fact, in order to make the various sets compatible, common subsets of input data are used to produce "reduced" versions of CT18, MSHT20 and NNPDF3.1 sets, which are then combined (now consistently, as they contain the same data) following specific statistical procedures. The authors shall change this sentence.

6) At page 12, above Eq. 3.12, the authors describe the calculation of the uncertainty from the choice of $\alpha_s$ at the Z pole. For better clarity, the authors could add that the additional eigenvector members of the PDF4LHC21 set indeed correspond to the choices $\alpha_s(M_Z)=0.117$ and $\alpha_s(M_Z)=0.119$, otherwise those numbers appear in the equation without an explanation.

7) At page 13, the authors describe the dependence of stau pair production cross section on the other SUSY paramters. The authors chose 10 TeV for the common mass of the light squarks, and 10.01 TeV for the second stop. This choice seems a bit fine tuned, and makes me think that there is somehow a dependence on the difference of these quantities. Could the authors explain the 0.01 TeV difference for these values?

8) At page 14, at the end of section 3.3, the authors symmetrise their uncertainty by changing the central value of the predictions in Eq. 3.18. I wonder if this is really necessary. Is there something wrong in having asymmetric error? More specifically I must admit that it pains me a bit to see that you change the result of the central value. That result comes from the PDF central value and in principle that represents the best value, given all the systematics. Is there another justification, apart from having symmetric errors, that I am missing for this choice of changing the central value?

9) This last point is just a comment about the comparison between predictions obtained with PDF4LHC15 and PDF4LHC21. As the Authors noted, the change of PDF set has a very large impact, often larger than the increase of the collisions' centre of mass energy, especially for heavy sparticles. This purely PDF driven effect shows how much PDF determination has changed during the past years, and it also reflects our current poor knowledge of PDFs at at high $Q^2$ and large Bjorken x (> 0.5-0.6). The difference between the two sets is largest when anti-quark PDFs are involved, as their densities at large x are poorly determined; processes with initial state gluons follow immediately, as gluon density evolution is directly coupled to the evolution of antiquark PDFs.

Recommendation

Ask for minor revision

  • validity: top
  • significance: high
  • originality: ok
  • clarity: high
  • formatting: perfect
  • grammar: excellent

Author:  Christoph Borschensky  on 2024-10-13  [id 4864]

(in reply to Report 1 on 2024-06-26)

We thank the referee for their positive evaluation of our work and their constructive comments. In the following, we reply to the points raised in the report, and list the corresponding changes in our paper:

  1. The $N$-independent two-loop contributions $\mathcal{C}^{(2)}_{ij\to kl,I}$ are indeed not yet known or calculated, and therefore not included in our results. The Coulomb factor contains terms originating from Coulomb-gluon exchange between the final-state particles, resummed up to all orders. This factor therefore contains terms of higher orders beyond one loop. To clarify that the hard-matching coefficient $\mathcal{C}^{(1)}_{ij\to kl,I}$ as mentioned in Eq. (3.6) only contains terms of one-loop order, we changed the last sentence of the paragraph:

    The terms that are non-logarithmic in $N$ from the NLO corrections, including one-loop virtual contributions, but excluding the Coulomb-gluon exchange between final states to avoid the double counting of these contributions, are collected by $\mathcal{C}_{ij\to kl, I}^{(1)}$.

  2. To clarify the sentence, we now write explicitly what is being subtracted:

    [...], and subtract the expanded from the resummed part.

  3. The effect of the boundstate contributions as compared to the NLO corrections and the resummed contributions is discussed in more detail in the arXiv:1607.07741 paper. In the current paper, we would prefer to focus on the updated results from NNLL-fast 2.0 and the differences with respect to the PDF sets and the centre-of-mass energy. We have however added a footnote to briefly mention the impact and size of the boundstate contributions:

    The boundstate contributions are generally positive and have a moderate effect on the total cross sections, leading to an increase with respect to NLO of a few per mille to the per cent range, as shown in [77 (arXiv:1607.07741)]. The effects are the largest for processes with large colour factors such as $\tilde g\tilde g$, and they become more relevant close to threshold, i.e.\ for smaller centre-of-mass energies or larger final-state masses, see also [70 (arXiv:1202.2260)].

  4. We moved the sentence below Eq. (3.7) starting with "The threshold-enhanced two-loop contribution [...]" now before Eq. (3.7), before the sentence starting with "This methodology ensures that [...]", so that the text now reads:

    Then, we add the fixed-order cross section $\sigma^{\mathrm{NNLO_{Approx}}}_{h_1 h_2 \to kl}$ of Eq. (3.1). We note that the threshold-enhanced two-loop contribution $\Delta\sigma_{h_1 h_2 \to kl}^{\mathrm{NNLO_{Approx}}}$ as included in $\sigma^{\mathrm{NNLO_{Approx}}}_{h_1 h_2 \to kl}$ differs from the corresponding term of the same order in the expansion of the resummed cross section by subleading contributions that are suppressed in Mellin space as $\mathcal{O}(1/N)$. The methodology of this matching procedure ensures that we combine the best known fixed-order result [...]

  5. We have corrected the corresponding sentence, which now reads:

    As a choice of parton densities, we make use of the recent PDF4LHC21 set [88], which combines the CT18 [89], MSHT20 [90], and NNPDF3.1 [91] global analyses.

  6. We added the information regarding the PDF members with $\alpha_s(M_Z) = 0.117$ and $\alpha_s(M_Z) = 0.119$:

    The PDF4LHC21 set includes additionally two members accounting for the 68% C. L. variation of $\alpha_s$ around its central value $\alpha_s(M_Z) = 0.118$, corresponding to the lower and upper values $\alpha_s(M_Z) = 0.117$ and $\alpha_s(M_Z) = 0.119$, within the determination procedure of the PDFs.

  7. The dependence of stop production on the additional parameters such as the light squark mass and the mass of the second stop, appearing only starting from NLO, is very small (see e.g. arXiv:1601.02954). Thus, we fix them to high values $\sim$10 TeV (which can then also be considered as a case where all particles other than the stop and the gluino are decoupled). The choice of having the difference of 0.01 TeV between the light squark and second stop masses is of numerical nature only, to avoid numerical divergences in the case of degeneracies. The results would be the same if the second stop mass were fixed to another value, e.g. 5 TeV or 15 TeV. For clarification in the paper, we added the following footnote:

    The difference of 0.01 TeV between $m_{\tilde q}$ and $m_{\tilde t_2}$ is only of computational nature to avoid numerical divergences in the degenerate case, and the actual choice of values for these masses has a negligible impact on the cross section, see also Table 3 of [76].

  8. We want to clarify that the results shown in the figures of the paper are plotted with asymmetric uncertainties around the central predictions according to Eq. (3.17) in the paper. Similarly, the output of NNLL-fast contains the central prediction and each theoretical uncertainty separately. The symmetrisation of the uncertainties around an average cross section value is only applied to several precomputed results for simplified scenarios on the TWiki of the LHC SUSY Cross Section Working Group. In order to clarify that the symmetric prescription is not used in the results presented in the paper, we removed Eq. (3.18) and the preceding sentence, and removed the text regarding the former Eq. (3.18) in the conclusions and added the following footnote:

    The predictions for coloured sparticles provided on the TWiki page are calculated as average cross sections $\sigma_{\text{avg}} := (U + L)/2$ with symmetric uncertainties $\delta_{\text{sym}} := (U - L)/(2\sigma_{\text{avg}}) = (U - L)/(U + L)$ with $U$ and $L$ as defined in Eq. (3.17), in agreement with the approach of [66].

  9. We thank the referee for the insightful comment.

---

## Round 1 · Referee Report · Anonymous (Referee 2) · 2024-7-8

Strengths

  1. The paper presents the best predictions for supersymmetric particle productions at the LHC.
  2. The paper is very well written and organised

Weaknesses

  1. In computing scale uncertainties, renormalisation and factorisation scales are the same. It is well-established to use a seven-point scale variation, with independent variation of renormalisation and factorisation scales by a factor of two, and avoiding the extremes where the ratio between the two is a factor 4.
  2. As far as I understand, the code provides only an interpolation of the provided grids. It would be useful to have the code that produces the points in the grids.

Report

The paper contains significant new results, setting the state of the art for the considered cross sections.

Requested changes

  1. Can the authors consider asymmetric scale variations? If there is a reason not to do so, this should be specified in the paper?
  2. Can the author clarify that the code does not allow the calculation of the points in the grid? Optionally, the authors might consider to make that code public as well.

Recommendation

Ask for minor revision

  • validity: high
  • significance: high
  • originality: good
  • clarity: high
  • formatting: excellent
  • grammar: excellent

Author:  Christoph Borschensky  on 2024-10-13  [id 4863]

(in reply to Report 2 on 2024-07-08)

We thank the referee for their positive evaluation of our work and their constructive comments. In the following, we reply to the points raised in the report, and list the corresponding changes in our paper:

  1. PROSPINO, the code used for the calculation of the NLO-(S)QCD cross sections, sets the renormalisation and factorisation scales equal, and it is not straightforward to disentangle the two. Calculating the 7-point scale uncertainty would therefore require either an intricate modification of the original PROSPINO code, or a new implementation of the squark and gluino processes at NLO, keeping the two scales separate from the beginning.

  2. NNLL-fast indeed only does an interpolation between pre-computed grid points. The actual computation of the grid points depends on two codes: PROSPINO for the NLO cross sections, and our own code for the resummed corrections. The merging of the data sets of both codes has to be performed properly and is not entirely straightforward, which is why we prefer to publish only the datasets where we already combined the fixed-order and resummed calculations. As a clarification, we rearranged the second paragraph of Section 4, such that we now start the paragraph with:

    The NNLL-fast code consists of pre-computed total cross sections and uncertainties provided as numerical grids, together with a fast interpolation code. All processes described in Sec. 2 are implemented. In addition to these processes, we also provide predictions for gluino-pair production in the limit of decoupled, i.e. very heavy, squarks, and squark-antisquark production in the limit of decoupled gluinos. The mass ranges of $m_{\tilde q/\tilde t_1}$ and $m_{\tilde g}$ for the grids are the following: [... Eqs. (4.1) to (4.4) ...] For the computation of the grid points at NNLO$_\text{Approx}$+NNLL accuracy, we employ the following codes. [...]}

    followed by the remaining text starting with "The NLO SUSY-QCD cross section is computed [...]".

---

## Round 1 · Referee Report · Anonymous (Referee 3) · 2024-7-16

Strengths

  1. Rather complete analysis of results at 13.6 TeV
  2. Fast interpolation code available

Weaknesses

  1. The physics content of the paper is limited, as the manuscript is mainly a collection of (updated) results at a higher c.o.m. energy and a different PDF set
  2. The code is not particularly flexible, as it can only work with a fixed c.o.m. energy (i.e. results at 14 TeV are not readily available) and a fixed PDF set

Report

The manuscript presents updated predictions at 13.6 TeV for pair production of coloured sparticle production. Most of the results are reproducible with the updated version of the NNLL-fast code which interpolates pre-tabulated grids using PDF4LHC21 NNLO parton densities.

Although the paper does not contain new material besides the updated predictions, and therefore the novelty aspects are limited, the availability and the documentation of these predictions are relevant for the discovery programme at the LHC Run 3 and I believe that the paper can be published in SciPost Physics.

Before recommending publication, I would ask the authors to address the following remarks.

  1. The theoretical treatment of the paper was discussed in the previous publication [77], and to my understanding has not been modified or improved. The results at higher accuracy are labelled interchangeably NNLO$_{\rm approx}$+NNLL or NNLL-fast. The labelling implies that the predictions contain, besides terms enhanced in the threshold region, additional terms at relative order $\alpha_s^2$ which are not captured by a fixed order expansion of the threshold resummed result. Nevertheless, according to the authors, the $\Delta \sigma_{\rm Approx}$ `collects the $\mathcal O(\alpha_s^4) $ contributions which are enhanced in the limit of sparticle pair-production taking place close to the threshold', and indeed the paper quoted describes the threshold expansion of sparticle pair production at $\mathcal O(\alpha_s^4) $. After Eq. (3.7), the authors indeed specify that the $\Delta \sigma_{\rm Approx}$ term differs from the corresponding term in the expanded results by subdominant $\Delta \mathcal O(1/N)$ terms. It seems to me that the final results does not contain any NNLO terms besides those enhanced at threshold, meaning that the formal accuracy of the calculation is equivalent to NLO+NNLL, as it does not contain any $\mathcal O(\alpha_s^4) $ term besides those included in the NNLL expansion.

  2. When producing the results at 13.6 TeV the authors comment on the impact of changing PDF and changing c.o.m. energy. The discussion highlights that the latter is particularly relevant. It would be interesting to include the PDF error for the ratio shown in Fig. 6, as the size of the PDF error can at present be inferred only indirectly through a comparison with Fig. 3 and 4. Given the relatively low originality of the results presented, perhaps adding some additional comment on the impact of the PDF uncertainty uncertainties at large masses would strengthen the need for up-to-date predictions in LHC searches.

  3. The squark-antisquark curve (dark green) curve in Fig. 9, top left panel has a peculiar behaviour between 2.5 and 3 TeV (which I reproduced with nnllfast-1.1) compared with the other curves and with its decoupled counterpart. Is there a reason for it?

  4. For consistency, resummed calculations should be computed using PDFs with threshold resummation effects. The only global set existent (1507.01006) contains only data from deep-inelastic scattering, Drell-Yan, and top quark pair and is relatively old, making it not competitive with more recent PDF analyses, albeit obtained using fixed order perturbation theory. Moreover, the effect of resummation was found to be minor at NNLO even at relatively large invariant masses. The authors could add a comment on this issue to justify the use of fixed order PDFs in their predictions.

  5. My last comment regards the distribution of the results. I appreciate the velocity and efficiency of interpolation grids, but it comes with limitations. Have the authors considered to release a version of the underlying code used to produce the grids?

Requested changes

  1. Please clarify the accuracy of the computation in the text. If the accuracy of the computation is indeed equivalent to a NLO+NNLL one (besides the addition of the bound states term) the use of the NNLO$_{\rm approx}$+NNLL label in the plots can be perceived as misleading. I would recommend the use of the label NNLL-fast in the plots, also to ensure a) consistency between plots and b) consistency with the label used in Eq. (3.7). The NNLL-fast label would also be more accurate as the inclusion of the bound states contribution would not be not transparent if the NNLO$_{\rm approx}$+NNLL (or NLO+NNLL) label is used.
  2. Please add PDF errors in Fig. 6.
  3. Please uniform $y$ range across Fig. 5, 6, 7, 8. Currently it is rather difficult to gauge quickly the relative effect of PDFs and c.o.m. energies as e.g. the upper right plot range changes from [0:5] in Fig. 5 to [$\sim$0.4:$\sim$3] in Fig. 6 to [$\sim$1.1:$\sim$1.7] in Fig. 7 to [0:6] in Fig. 8.
  4. Please add a label in Fig. 9 (or modify the caption) to clarify when NLO PDFs are used for NLO predictions to make the figure self-consistent. Check behaviour of squark-antisquark cross section with PDF4LHC15.
  5. Consider whether to comment on the use of resummed PDFs.
  6. In the conclusion, I believe that a slight rephrasing when comparing to NNLL-fast 1.1 could be beneficial to highlight the differences between 2.0 and 1.1. Rather than `In comparison to the previous version 1.1 of NNLL-fast, we found that ...', it would perhaps better to state more explicitly the differences with respect with the old predictions, for instance: 'The NNLL-fast 2.0 predictions supersede those obtained with NNLL-fast 1.1, which were computed at 13 TeV using the PDF4LHC 15 PDF set. By comparing the predictions, we found...'.

Recommendation

Ask for minor revision

  • validity: good
  • significance: ok
  • originality: low
  • clarity: high
  • formatting: excellent
  • grammar: excellent

Author:  Christoph Borschensky  on 2024-10-13  [id 4862]

(in reply to Report 3 on 2024-07-16)

Warnings issued while processing user-supplied markup:

  • Inconsistency: Markdown and reStructuredText syntaxes are mixed. Markdown will be used.
    Add "#coerce:reST" or "#coerce:plain" as the first line of your text to force reStructuredText or no markup.
    You may also contact the helpdesk if the formatting is incorrect and you are unable to edit your text.

We thank the referee for their positive evaluation of our work and their constructive comments. In the following, we reply to the points raised in the report, and list the corresponding changes in our paper. First, the comments listed in the "Report":

  1. The NNLO$_\text{Approx}$ indeed only contains threshold-enhanced terms at $\mathcal{O}(\alpha_s^4)$, i.e. terms proportional to powers of $1/\beta$ (from Coulomb contributions) as well as $\ln\beta$ (from soft-gluon contributions as well as non-Coulomb relativistic kinetic-energy corrections). However, whether to do NLO+NNLL or NNLO$_\text{Approx}$+NNLL matching, i.e. whether to include the $\mathcal{O}(\alpha_s^4)$ terms through $\Delta\sigma^{\text{NNLO}_\text{Approx}}$ or through the expansion of the resummed NNLL exponent, can lead to differences of $\mathcal{O}(10\%)$ originating from the subleading $\mathcal{O}(1/N)$ terms, as shown e.g. for stop-pair production in arXiv:1601.02954, Fig. 7. This difference due to the subleading contributions is also pointed out in the paper at the end of Section 3.2. We therefore prefer to keep the distinction between NLO+NNLL and NNLO$_\text{Approx}$+NNLL, the latter which we use for our predictions. The labels in the figures have been changed from "NNLO$_\text{Approx}$+NNLL" to "NNLL-fast", as suggested by the referee.

  2. We have added PDF+$\alpha_s$ uncertainties to Fig. 6, added the following text to its caption:

    The shaded bands denote the size of the PDF+$\alpha_s$ uncertainties from the PDF4LHC21 set around the central cross section predictions in the numerator, in the same colour scheme as the lines.

    and added the following text to discuss the impact of these uncertainties:

    Additionally, we show in Fig. 6 the size of the PDF+$\alpha_s$ uncertainties plotted around the central cross section predictions in the numerator, which are calculated with the PDF4LHC21 set. While the effect of updating the PDF set from PDF4LHC15 to PDF4LHC21 is, for the shown mass range, contained entirely within the uncertainty bands, the uncertainties grow in particular for the high-mass region to very large values, which highlights the need for a precise determination of PDFs in the large-$x$ region in order to properly constrain the squark and gluino processes at large $m_{\tilde q}$ and $m_{\tilde g}$.

  3. The $K$-factor for squark-antisquark production with the PDF4LHC15 set has been discussed in arXiv:1607.07741 around Fig. 3. The behaviour originates from a property of the PDF4LHC15 set (using Monte Carlo combination), leading to negative cross sections for several PDF replicas for high squark and gluino masses above ~2 TeV. For the 13 TeV results, we applied the prescription of setting the negative cross sections for the specific replicas to zero before evaluating the central cross section prediction. The occurrence of negative cross sections was particularly pronounced for squark-antisquark production in the high-mass region, resulting in the observed behaviour of the $K$-factor. Its decoupled counterpart differs in the dependence on the particle luminosities: while squark-antisquark production is dominated by quark-antiquark luminosities through the tree-level diagram of $t$-channel gluino exchange (see Fig. 1b, second diagram, in this paper), this diagram is absent in the case of decoupled gluinos. Thus, the decoupled case depends more strongly on the gluon luminosities, which, in our case, resulted in fewer or no replicas leading to negative cross sections.

    We have added a comment regarding the behaviour of the $K$-factor for squark-antisquark production to the discussion in Section 5.3:

    The behaviour of the $K$-factor for $\tilde q\tilde q^*$ production at $\sqrt{S} = 13$ TeV for masses above 2.5 GeV in the upper left panel of Fig. 9 has been discussed in [77]. It is due to setting negative cross sections as obtained for several replicas of the PDF4LHC15 set to zero to obtain a positive central cross section prediction, and the effect is most pronounced for the case of $\tilde q\tilde q^*$ production.

  4. The effect of using PDF sets with threshold resummation effects for squark and gluino production at NLO+NLL was studied in arXiv:1510.00375, where we found that the shift induced by the resummation-improved PDF was contained within the total PDF uncertainty band of using conventional PDFs. We added a comment to the end of the first paragraph of Section 5:

    As an additional remark, we note that our calculation including threshold resummation would in principle require using a threshold-improved PDF set, such as the one from [94 (arXiv:1507.01006)], for numerical predictions. However, as there is no threshold-improved PDF set based on more recent data, we consider using modern sets such as PDF4LHC21, albeit determined on the basis of only fixed-order predictions, preferable. In [67 (arXiv:1510.00375)], the effect of the PDF set from [94 (arXiv:1507.01006)] on squark and gluino production was studied at NLO+NLL accuracy, and it was found that the difference between a conventional fixed-order PDF set and the one including threshold resummation effects is contained within the total PDF uncertainty of the conventional set.

  5. The computation of the interpolation grid depends on two codes: PROSPINO for the NLO cross sections, and our own code for the resummed corrections. The merging of the data sets of both codes has to be performed properly and is not entirely straightforward, which is why we prefer to publish only the datasets where we already combined the fixed-order and resummed calculations.

Then, we reply to the "Requested changes":

  1. See reply "1." above.

  2. See reply "2." above.

  3. We prefer to keep the different $y$ ranges for the plots in Figs. 5 to 8, as otherwise, we would then have to set the plots to the largest $y$ range, which would make it difficult to distinguish the lines in the plots with a smaller range of $y$ values.

  4. We have added a sentence to the caption of Fig. 9 to clarify that, whenever NLO PDFs are available for a specific PDF set, they are used for NLO cross sections:

    Whenever NLO PDFs are available, i.e. in the case of the PDF4LHC15 set, we use NLO PDFs for the calculation of $\sigma^{\text{NLO}}$ in the denominator. In all other cases, we use NNLO PDFs.

  5. See reply "4." above.

  6. We added a sentence to highlight more explicitly the difference to the previous NNLL-fast 1.1, as suggested by the referee:

    The NNLL-fast 2.0 predictions supersede those obtained with NNLL-fast 1.1, which were computed at 13 TeV using the PDF4LHC15 set. By comparing the predictions, we found that the total cross sections increased uniformly by up to about 60-70% in the probed mass regions for all processes by changing the centre-of-mass energy from 13 TeV to 13.6 TeV. The update from PDF4LHC15 to the newer PDF4LHC21 set [...]

---

## Editorial Decision

resubmitted